# Therapeutic efficacy of dimethyl fumarate in relapsing-remitting multiple sclerosis associates with ROS pathway in monocytes

Karl E. Carlström[1], Ewoud Ewing [1], Mathias Granqvist[1,8], Alexandra Gyllenberg[1,8], Shahin Aeinehband[1], Sara Lind Enoksson[2], Antonio Checa[3], Tejaswi V.S. Badam[4,5], Jesse Huang[1], David Gomez-Cabrero[6], Mika Gustafsson [7], Faiez Al Nimer[1], Craig E. Wheelock [3], Ingrid Kockum[1], Tomas Olsson[1], Maja Jagodic[1] & Fredrik Piehl [1]

Dimethyl fumarate (DMF) is a first-line-treatment for relapsing-remitting multiple sclerosis (RRMS). The redox master regulator Nrf2, essential for redox balance, is a target of DMF, but its precise therapeutic mechanisms of action remain elusive. Here we show impact of DMF on circulating monocytes and T cells in a prospective longitudinal RRMS patient cohort. DMF increases the level of oxidized isoprostanes in peripheral blood. Other observed changes, including methylome and transcriptome profiles, occur in monocytes prior to T cells. Importantly, monocyte counts and monocytic ROS increase following DMF and distinguish patients with beneficial treatment-response from non-responders. A single nucleotide polymorphism in the ROS-generating *NOX3* gene is associated with beneficial DMF treatment-response. Our data implicate monocyte-derived oxidative processes in autoimmune diseases and their treatment, and identify *NOX3* genetic variant, monocyte counts and redox state as parameters potentially useful to inform clinical decisions on DMF therapy of RRMS.

[1] Department of Clinical Neurosciences, Section of Neurology, Karolinska Institutet, Stockholm, Sweden. [2] Department of Clinical Immunology Karolinska University Hospital, Stockholm, Sweden. [3] Division of Physiological Chemistry II, Department of Medical Biochemistry and Biophysics, Karolinska Institutet, Stockholm, Sweden. [4] Department of Bioinformatics, School of Bioscience, University of Skövde, Skövde, Sweden. [5] Department of Physics, Chemistry & Biology (IFM), Bioinformatics, Linköping University, Linköping, Sweden. [6] Translational Bioinformatics Unit, Navarrabiomed, Complejo Hospitalario de Navarra (CHN), Universidad Publica de Nevarra (UPNA), IdiSNA, Pamplona, Spain. [7] Department of Physics, Chemistry and Biology, Linköping University, Linköping, Sweden. [8] These authors contributed equally: Mathia Granqvist, Alexandra Gyllenberg. Correspondence and requests for materials should be addressed to K.E.C. (email: karl.carlstrom@ki.se)

An increasing body of evidence suggests that redox reactions are important for the regulation of immune responses during infection, malignancies and autoimmunity[1]. Relapsing-remitting multiple sclerosis (RRMS) is an autoimmune disease associated with dysregulation of adaptive immunity, leading to the periodic entry of immune cells into the central nervous system (CNS) and subsequent tissue damage with symptoms of neurological dysfunction. Among a number of different pathological disease mechanisms, an imbalance in the oxidative environment has also been described[2,3].

Dimethyl fumarate (DMF; Tecfidera®) is a oral disease modulating treatment (DMT) and the most prescribed drug for RRMS in the U.S. It's suggested to act by activating the transcription factor nuclear factor (erythroid-derived 2)-like 2 (Nrf2)[4,5], which is a transcript of the NFE2L2 gene. Nrf2 is essential in redox homeostasis and responses to reactive oxygen species (ROS)[1] but may in addition also engage additional transcription factors, including NFκB[6]. The net activity of DMF has been described to be mainly anti-oxidative[5,7,8]. So far targeting of redox regulation has not been a generally accepted therapeutic strategy in autoimmune diseases. However, older drugs including gold salts used for rheumatoid arthritis have been shown to possess redox regulatory properties[9–11]. DMF has been ascribed cyto-protective effects of potential relevance for CNS cells during inflammation, but conclusive data on degree of CNS penetration in humans is still lacking[12,13] and modulation of disease relevant T cell subsets[14–19], therefore remains the most likely mechanism for reducing clinical and neuroradiological disease activity in RRMS[5,20,21]. Hence, we here chose to assess the effects of DMF in peripheral blood.

Experimental evidence establish ROS as potent immune regulators, suggesting that dampening of oxidative reactions paradoxically may increase susceptibility to autoimmune diseases[22–24]. Thus, a naturally occurring genetic variant in the rat Ncf1 gene, encoding a ROS-generating NADPH oxidase subunit, leading to lowered ROS generation is associated with increased susceptibility to both experimental autoimmune arthritis[25] and encephalomyelitis (EAE)[22,26], the animal model for MS. Experiments in genetically modified mice have pin-pointed this effect to incapacity of myeloid cells to limit T cell proliferation and to induce regulatory T cell (T$_{reg}$) activation via superoxide generation[27–29]. ROS has also been shown to mediate a range of immune regulatory effects including; T cell hyporesponsiveness[30,31], diminished T cell receptor signaling[22,32–34], cytokine production[35] and T helper cell (T$_H$17) development[36–38]. In addition, memory T cells are more susceptible to ROS compared to naïve T cells and T$_{reg}$[39]. Collectively, these observations suggest that ROS can modulate multiple immune responses of importance in autoimmunity. Monocytes are potent producers of ROS primarily via NADPH oxidases[40,41]. In mice monocytes have been shown to regulate disease models of autoimmunity[31,38,42], and ROS deficiency causes failure of this regulation[43,44]. Still, the existing literature mostly consists of experimental animal or in vitro studies, and studies performed ex vivo in man during drug interventions are rare. Regulation of redox reactions and oxidative damage within the CNS is relevant in a range of conditions, including MS, where signs of oxidative damage and expression of anti-oxidative proteins are found in active MS lesions[45,46]. This study, however, was restricted to a detailed characterization of the initial monocyte response and subsequent immunomodulation occurring in peripheral blood of RRMS patients starting therapy with DMF. In addition, DNA methylation changes in sorted cells were used to verify changes in transcription and immunoprofiling as well as to provide additional relevant mechanistic informaton given the emerging role of DNA methylation in regulating immune response and inflammatory diseases[47,48].

Herein, we identify DMF to increase monocyte ROS generation and that epigenetic methylation changes in monocytes precede those occurring in CD4$^+$ T cells. Furthermore, a reduced capacity to generate ROS and lower monocyte counts is associated with reduced clinical efficacy of DMF. Lastly, we identify a single nucleotide polymorphism (SNP) in the NOX3 gene to be associated to a beneficial treatment response to DMF and suggestively associated with increased ROS generation.

## Results

**Early monocyte response to DMF reveals therapy efficiency.** DMF has been included in the Swedish public reimbursement program for treatment of RRMS since May 2014. We included patients from May 2014 to March 2017 starting DMF in clinical routine that volunteered for extra blood sampling, but otherwise were not subject to any other selection criteria (Fig. 1a–c, Supplementary Dataset 1). Peripheral blood was collected at regular intervals before (baseline) and during the first six months after starting DMF and patients were followed in order to evaluate treatment efficacy according to clinical routine. To address oxidative stress, we initially determined plasma levels of free 8.12-iso-iPF2α-VI isoprostane generated through non-enzymatic oxidation, considered as the most acknowledged technique of quantifying oxidative stress[49,50] and superior to measurement of e.g., anti-oxidative enzymes since this technique measure oxidation instead of secondary responses to oxidation. Isoprostane 8.12-iso-iPF2α-VI was significantly increased compared to baseline levels already three months following DMF treatment, this effect was sustained after six months, suggesting an increase in oxidative environment (Fig. 1d). This could be observed at the transcriptional levels, as Gene Set Enrichment Analysis (GSEA) on differentially expressed mRNAs in CD14$^+$ monocytes at baseline and after six months showed an enrichment of upregulated genes involved in response to oxidative stress as compared to baseline (including TXN, SOD1/2, NFE2L2) (Fig. 1e, f). In addition, unbiased Ingenuity Pathway Analysis (IPA) demonstrated altenation of Nrf2, NFκB, HIF1α and fatty acid oxidationpathways in response to DMF (Fig. 1g, Supplementary Dataset 2 Supplementary Table 1).

Depending on the treatment outcome, patients were either categorized as DMF responders if they had continuous DMF therapy for at least 24 months without signs of disease activity (i.e., free of clinical relapses and newly appearing brain lesions on magnetic resonance imaging (MRI), or DMF non-responders if they displayed signs of continued clinical and/or neuroradiological disease activity at any stage after the first three months (Fig. 1b). At three months, responders had significantly higher counts of CD14$^{++}$CD16$^-$ cells, representing the main monocyte population, compared to non-responders, while the remaining monocyte subsets did not differ between the two groups or over time (Fig. 2a–d). A difference in total monocyte counts between responders and non-responders was subsequently replicated using retrospective data for a larger cohort of RRMS patients starting DMF (Fig. 2e, f). Subjects in Fig. 2e, f were not included in other cellular immune profiling experiments. Detailed description of all subject can be found in Fig. 1c and Supplementary Dataset 1. Furthermore, early changes in monocyte counts at three months were negatively associated with changes in lymphocyte counts at 12 months. Hence, non-responders displayed lower monocyte counts at the earlier time point and higher lymphocyte counts at 12 months compared to responders (Fig. 2f).

Next, we determined monocytic ROS generation in RRMS patients and controls using dihydrorhodamine-123 (DHR-123).

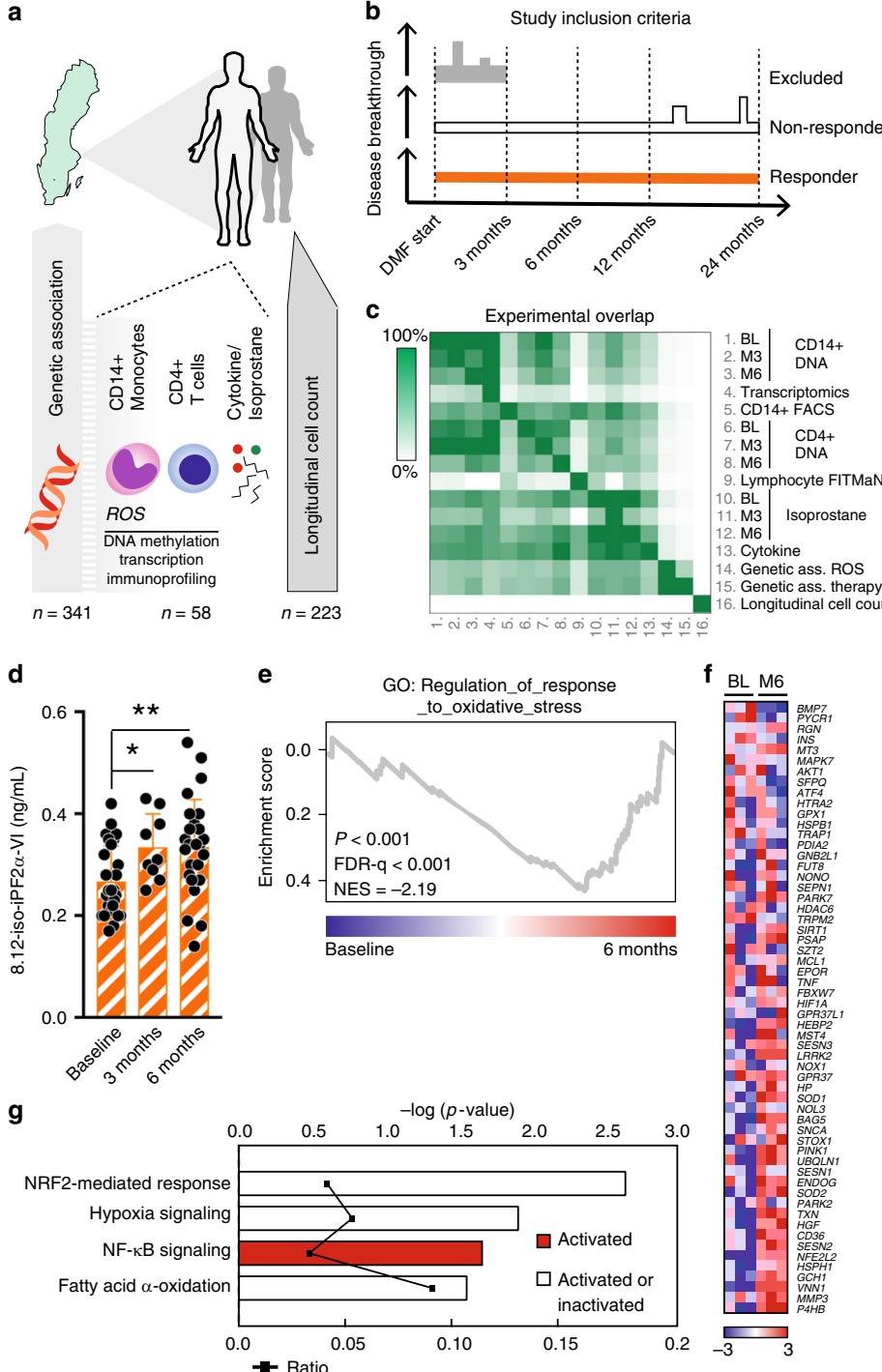

**Fig. 1** DMF induce increased isoprostane oxidation and transcription in response to oxidative stress. **a** Scheme of the study design. Peripheral blood was sampled at the clinic of patients fulfilling the criteria for RRMS and prescription of DMF. **b** Definition of RRMS patients as responders or non-responders to DMF therapy. **c** Experimental overlap between different assessments in the study **d** 8.12-iso-iPF2α-VI-isoprostane in plasma from paired patients sampled at three months ($n = 9$) and six months ($n = 26$) after DMF start compared to baseline ($n = 26$). **e, f** GSEA on mRNA from CD14+ sorted monocytes at baseline ($n = 3$) and six months ($n = 3$) shows an enrichment of upregulated genes involved in GO_REGULATION_OF_RESPONSE_TO_OXIDATIVE_STRESS at six months. ES, $P$-value and FDR were calculated by GSEA with weighted enrichment statistics and ratio of classes for the metric as input parameters. **g** Pathway analysis demonstrates oxidative stress-related canonical pathways in response to DMF ($n = 3$). Differentially expressed genes ($P < 0.01$ with average RMA > 4, one-way ANOVA) were used in IPA and significant pathways were determined with the right-tailed Fisher's exact test ($P < 0.05$). The ratio indicates pathway's activation status upon DMF treatment (**g**). Analysis in (**d**) was done by one-way ANOVA comparing both time points to baseline and graph shows mean and S.D. *$P < 0.05$, **$P < 0.01$

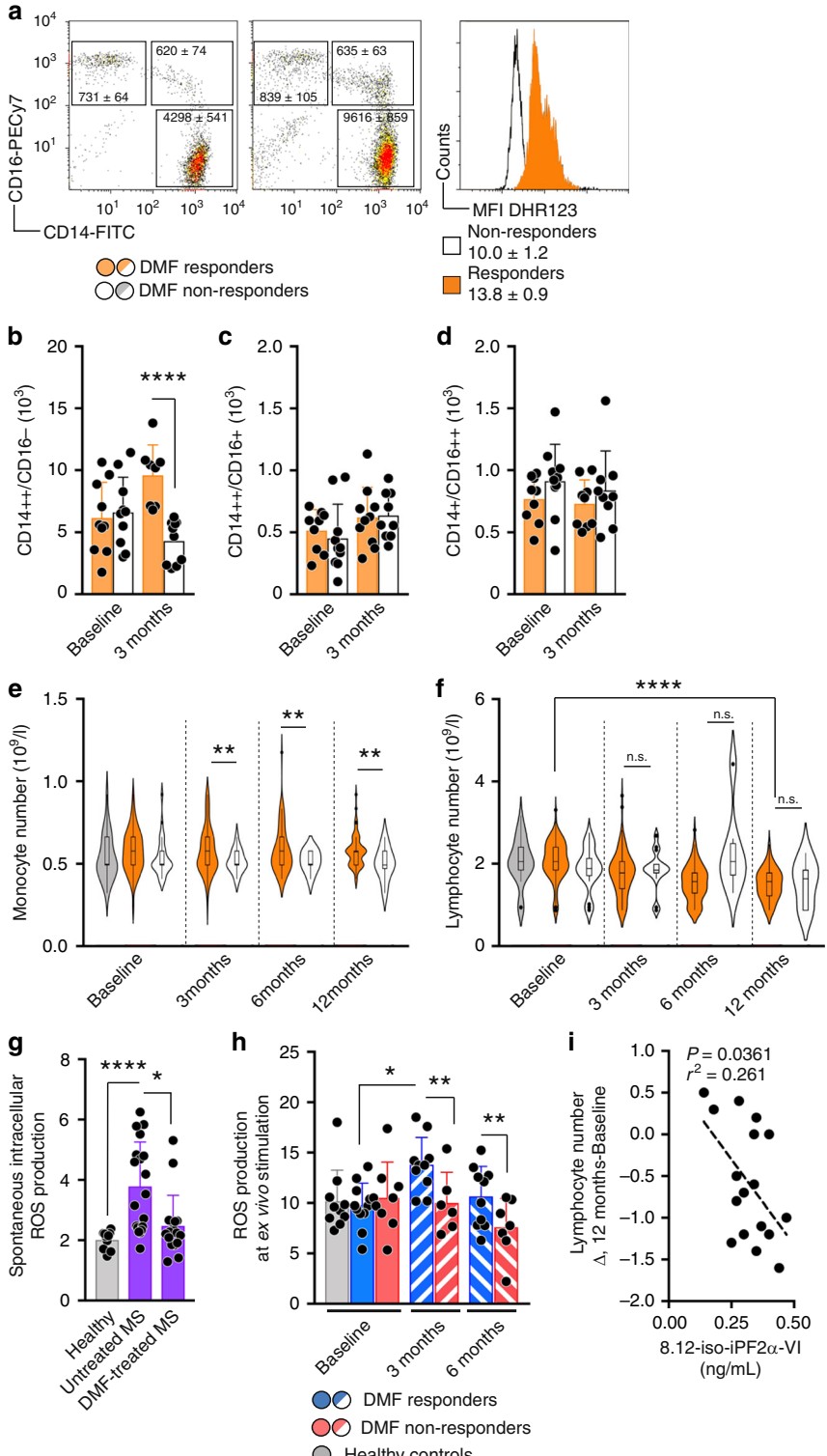

**Fig. 2** Monocyte numbers and ROS generation separate DMF responders from non-responders. **a** Representative plots for gating strategy of monocyte subsets stating mean ± S.E.M. at three months. Histogram shows representative DHR-123 MFI ± S.E.M. at 3 months. **b**–**d** Number of monocyte subsets in DMF responders ($n = 10$) and non-responders ($n = 10$) at baseline and at three months. **e** Violin with box-and-whisker plots indicating values outside the 5–95 percentile are indicated as dots of monocytes and (**f**) lymphocytes from healthy controls ($n = 28$), DMF responders ($n = 171$) and DMF non-responders ($n = 26$) over time. **g** Spontaneous generation of ROS in healthy controls ($n = 10$) and DMF untreated ($n = 18$) and treated ($n = 18$) patients. **h** ROS generation in *ex vivo*-stimulated monocytes from DMF responders ($n = 10$) and non-responders ($n = 7$) measured with DHR-123. **i** Correlation between Δlymphocyte number and 8.12-iso-iPF2α-VI-isoprostanes at 6 months determined with Spearman's correlation ($n = 17$). Graph (**b**–**d**, **g**, **h**) shows mean and S.D. Analysis in (**f**) was performed with ANOVA for linear trend over time and (**e**) to test mean comparison between responders and non-responders at every timepoint. Remaining analysis between paired patients were performed with paired *t* test whereas analysis between healthy controls and patients or between patient groups were performed with Student's two-tailed *t* test. *$P < 0.05$, **$P < 0.01$, ***$P < 0.001$, ****$P < 0.0001$

Baseline levels of spontaneous ROS generation before starting DMF, although very low, was higher in RRMS patients than healthy controls, DMF treatment resulted in decreased spontaneous ROS generation independently of later being a responder or non-responder (Fig. 2g). In contrast, responders displayed a more vigorous increase in ROS generation upon ex vivo stimulation with *E. coli* compared to non-responders at both three and six months (Fig. 2h). ROS generation was not altered in lymphocytes or granulocytes over time (Supplementary Fig. 1). These findings, together with a correlation observed between isoprostane levels at six months and degree of reduction in lymphocyte counts from baseline to 12 months, suggest a link between the ROS response and subsequent changes to the peripheral lymphocyte compartment (Fig. 2i).

**DNA methylome changes occur early in monocytes**. To provide more insight into the molecular mechanisms involved in DMF effect, we profiled DNA methylation, a stable epigenetic mark able to influence transcriptional activity, immune functions and disease[47,48] using Illumina EPIC arrays in sorted CD14$^+$ monocytes from RRMS patients sampled at baseline and after three and six months. Assessment of other types of epigenetic changes, such histone modifications and non-coding RNAs was not performed due to limitations in input material. Monocytes displayed numerous methylation changes, but none reaching false discovery rate (FDR) significance, possibly due to the low number of analyzed samples (Fig. 3a, Supplementary Dataset 3). The most pronounced methylation changes occurred between baseline and three months which then reverted back to baseline levels after six months of treatment (Fig. 3b). Thus, methylation patterns revealed three month as a critical time window, with most changes being identified in baseline vs. three months and three months vs. six months, while very little change has been observed between baseline and six months. Pathways associated with differentially methylated genes implicate functions related to regulation of apoptosis (*PRKCZ/B, INPP5D*), metabolism (*IRS2 NR1H3*), cell communication (*IL6, STAT3*) and migration (*NFAT, IL6*), (Fig. 3c, Supplementary Table 2). A substantial number of the pathways also contained genes known to respond to ROS as defined by GO_RESPONSE_TO_OXIDATIVE_STRESS, depicted in the Fig. 3c by increasing blue color. Further, we tested whether methylation levels in monocytes at baseline correlated with the response to DMF in 11 responders and three non-responders. We found three highly suggestive proximal CpGs mapping to a locus on chromosome 16, cg04536393 (adj.*P*-val < 0.090, 6.8% lower methylation in responders), cg27075654 (adj.*P* val < 0.065, 23.6% higher methylation in responders) and cg07622957 (adj. *P* val < 0.065, 30.1% higher methylation in responders) associating to treatment response. While the latter two CpGs map to a region in the *CLCN7* gene predicted to be transcribed, cg04536393 maps to an intergenic region annotated to be an enhancer in monocytes[51]. Although of potential interest, additional studies in larger cohorts will be needed to explore if these locus harbors gene(s) which methylation status is of relevance for the DMF treatment outcome.

**NOX3 SNPs associates with DMF-therapy outcome**. In order to address any possible genetic contribution to ROS generation in monocytes and response to DMF treatment, i.e., supporting a causative role, we analyzed a set of SNPs in genes encoding some of the components of the NADPH oxidase 1–4 complexes. One SNP in *NOX3* (rs6919626) displayed suggestive association

($\beta = -0.28$; $P = 0.057$), with the minor *G* allele contributing to reduced ROS generation in monocytes after ex vivo stimulation with *E. coli* (Fig. 4a, Supplementary Dataset 5). Notably, the same allele was also significantly associated (OR = 1.57; $P = 0.036$) with likelihood of displaying an insufficient response to DMF (Fig. 4a, Supplementary Dataset 5). Several additional SNPs within *NOX3*, *NOXO1* and *CYBA* showed significant association with response to DMF treatment (Supplementary Dataset 5), however, in no case the same marker displayed association both to ROS generation and DMF treatment response. *NOX3* is not solely expressed in monocytes. To further propose a mechanistic rational for genetic variations in rs6919626 in association to ROS and therapy outcome, we assessed methylation in the *NOX3* promoter region and expression of *NOX3* in sorted CD14$^+$ monocytes (Fig. 4b, c). The *A* allele that associated with the response to DMF exhibited consistent tendency for association with lower methylation at several CpGs in the promoter of *NOX3*, already at baseline (Fig. 4b, c). Reduced CpG methylation in DMF responders carrying the *A* allele could be further linked to higher *NOX3* transcription in CD14$^+$ monocytes at six months (Fig. 4d). Together, this finding suggests that genetic variation and CpG methylation in monocytic *NOX3* might influence the *NOX3* gene transcription and thus monocyte function, particularly ROS production.

**Delayed changes in CD4$^+$ T cells after DMF intervention**. We next investigated DNA methylation in CD4$^+$ T cells and identified numerous significant methylation changes following treatment with DMF (Fig. 5a). In contrast to monocytes, methylation changes in CD4$^+$ T cell mostly occurred between three and six months after DMF start, compared to baseline versus three months (Fig. 5a, b, Supplementary Dataset 6). Altered CpGs displayed predominant hyper-methylation at genes involved canonical pathways related to T cell differentiation (especially T$_H$17 and T$_H$17/T$_{reg}$ balance), migration, development and apoptosis (Fig. 5c, Supplementary Table 3). In addition, these clusters were also affected by ROS based on GO_RESPONSE_TO_OXIDATIVE_STRESS. Pathway analyses performed on genes associated to methylation changes in CD4$^+$ T cells between three and six months of DMF treatment implicate regulation of proliferation and apoptosis, migration, differentiation of T$_H$17 and T$_{reg}$ (Supplementary Table 4). Significant upstream regulators predicted to explain the DNA methylation changes include transcriptional regulators important in T cell activation and T$_{reg}$ function (*EP300/CREBBP*) as well as T$_H$17/T$_{reg}$ balance (*IL17A, RORC, STAT5B*) (Supplementary Table 5). To further explore T cells response to DMF, we conducted longitudinal characterization of T cell subsets and plasma cytokine levels using multicolor flow cytometry and Olink platform, respectively, at baseline and at six months following DMF treatment start. We found a significant increase in proportions of naïve T cells in responders over time, compared to baseline as well as, compared to non-responders at six months (Fig. 6a), while total naïve T cell numbers decreased in both groups over time (Fig. 6b). Proportions and absolute numbers of central memory T cell (T$_{CM}$) and effector memory T cells (T$_{EM}$) were significantly lower in responders over time (Fig. 6c–f) and unchanged in non-responders. Unlike naïve T cell subset, changes in absolute numbers of T$_{CM}$ and T$_{EM}$ also followed the same direction as relative cell counts (Fig. 6d, f). Difference in cell proportions between responders and non-responders was accompanied by more pronounced changes in cytokine profiles over time in responders compared to non-responders (Fig. 6g, h). For example levels of IL-17C, IL-12B and CXCL9

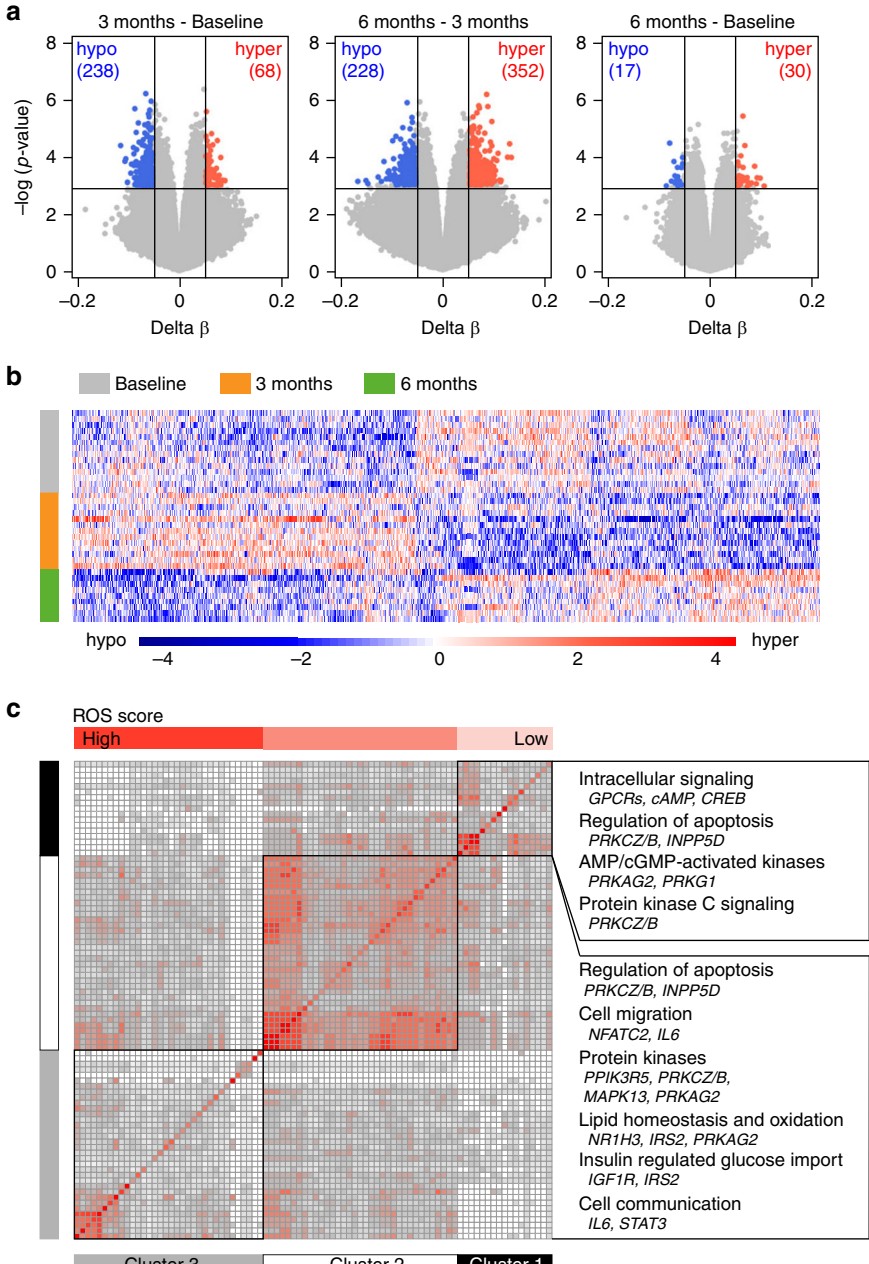

**Fig. 3** Monocytes undergo DNA methylation changes at three months. **a** DNA methylation was measured using Illumina EPIC arrays in CD14⁺ monocytes sorted from peripheral blood of RRMS patients before (baseline; BL, $n = 14$) treatment with DMF as well as three and six months following the treatment (M3 $n = 12$ and M6 $n = 9$). Volcano plots illustrating differences in DNA methylation between different time points following DMF treatment. Hyper- and hypo-methylated CpGs with min 5% methylation change and $P < 0.001$ (Linear model testing) are indicated in red and blue, respectively. **b** Heat map of 1614 most significant differentially methylated CpG sites between the time points (the scale represents $z$-score). **c** Clusters of canonical pathways, derived using Ingenuity Pathway Analysis, suggest functions that are affected in monocytes following DMF treatment. The significance of pathways was determined with the right-tailed Fisher's exact test ($P < 0.05$) and the clustering was performed on the relative risk for the overlap of molecules between the pathways using k-means. The horizontal top bar indicates the degree of overlap with selected ROS genes. Summary of the key functions and their constituent genes that displayed changes in DNA methylation upon DMF treatment are given to the right

were significantly lower in responders over time but not in non-responders over time (Supplementary Table 6). Altogether, several pathways in CD4⁺ T cells are being epigenetically changed and contains genes being affected by ROS. Lowering of $T_{CM}$ and $T_{EM}$ is more pronounced in responders compared to non-responders to DMF. While the largest difference in cytokine profile is present over time but less evident between responders and non-responders.

## Discussion

Experimental and clinical data suggest redox reactions to be involved in regulation of immune reactions in autoimmune conditions[24,52,53]. For example, non-phagocytic ROS has been shown to regulate autoreactive T cells both in models of arthritis[25] and MS-like disease[22]. However, evidence supporting a disease regulatory role of ROS generation in human studies including therapeutic intervention is overall rare and so far

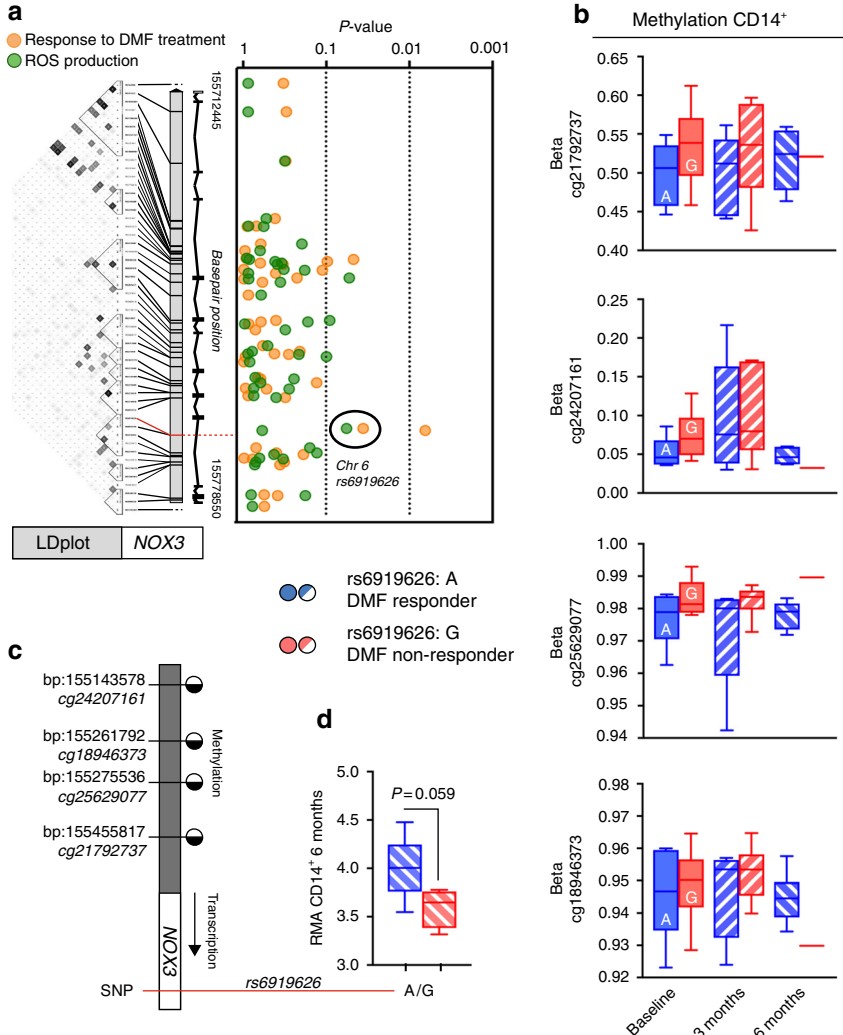

**Fig. 4** Genetic association with SNPs in *NOX3* to ROS generation and response to DMF treatment. **a** The *G* allele of rs6919626 (*red line*) was suggestively associated with reduced ROS generation in monocytes after ex vivo stimulation with *E. coli* ($P = 0.057$, $\beta = -0.28$) and significantly associated with lack of response to DMF treatment ($P = 0.036$, OR = 1.57). The Linkage Disequilibrium (LDplot) of the markers in *NOX3* was generated with HaploView4.2 in the Swedish population. Darker gray indicates higher $r2$ between markers. **b** Methylation in four CpGs in the *NOX3* promotor region over time and between responders (*A* allele) and non-responders (*G* allele) (Baseline: $n = 6 + 6$, three months: $n = 5 + 6$, six months: $n = 6 + 1$). **c** Schematic illustration indicating methylated or unmethylated CpGs in the *NOX3* promotor region. **d** *NOX3* transcription in responders and non-responders at six months following DMF ($n = 5 + 4$). Analysis performed using Welch's two-tailed *t* test. Graph (**b**, **d**) shows box-and-whisker plots indicating values outside the 5–95 percentile are indicated as dots

lacking in MS. The way by which DMF is beneficial to RRMS patients stands out from other currently used DMTs, since its mode of action cannot readily be explained by a direct effect on the lymphocyte compartment and may include effects on multiple cell types and signaling pathways[54], notably through its action on Nrf2, and to some degree other transcription factors. Nevertheless, a substantial body of evidence shows that T cell functions, believed to be important for MS disease pathogenesis, are indeed modulated by DMF[14,18]. In addition, clinical effects are slow and become noticeable after the first three months of treatment, as observed already in the early trial program[20,21]. We herein provide insights into early effects of DMF on monocytes that could be of importance for subsequent modulation of adaptive immune responses. So far myeloid cell-derived ROS generation as an immune regulator has received little attention during DMT intervention in RRMS, including DMF. This is despite extensive experimental data supporting a role for ROS in the regulation of adaptive immune responses in autoimmune conditions[28,36]. In

light of the anti-oxidative features ascribed to DMF-mediated-Nrf2-activation[1,4], we herein found DMF to contribute to an net increase of the oxidative environment, as shown by increased free isoprostane level produced by non-enzymatic oxidation (Fig. 1). Previous studies have indicated similar changes based on less reliable indirect measurements, such as increased transcription or anti-oxidative protein depletion[5,17,19].

Importantly, monocytes from patients without signs of clinical or neuroradiological disease breakthrough had an increased capacity to produce ROS compared to patients that had such signs. Monocytes are potent producers of ROS through their NADPH oxidase complexes, but in contrast to other ROS producing cells, they are generally not considered to play a major role in e.g., microbial defense. They have thus been suggested to have other functions, including immune regulation. While alterations in monocyte subsets between healthy individuals and autoimmune patients, including MS, have been observed previously, longitudinal studies during immune intervention are scarce.

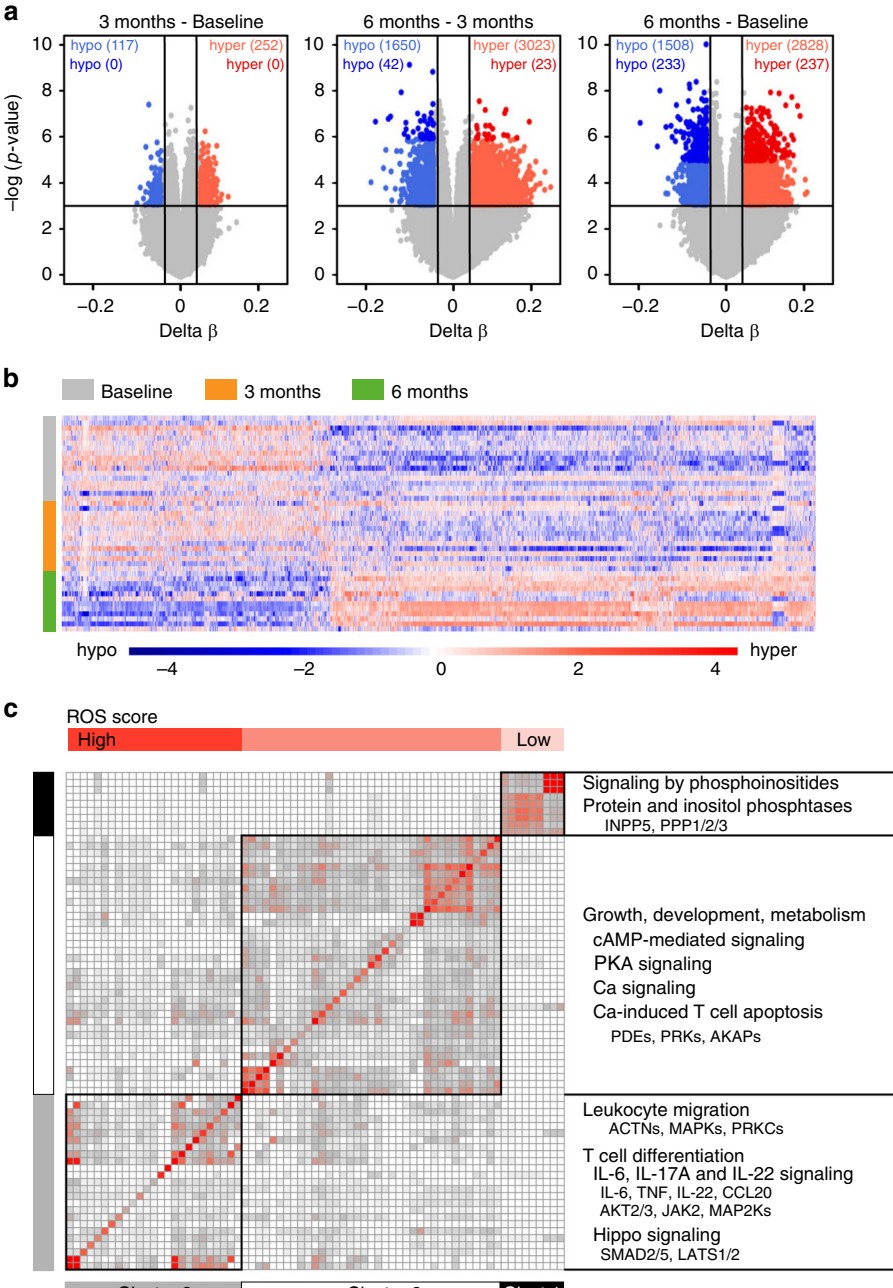

**Fig. 5** DNA methylation changes are delayed in T cells and occur after six months. **a** DNA methylation was measured using Illumina EPIC arrays in CD4[+] T cells sorted from peripheral blood of RRMS patients before (baseline; BL, $n = 17$) treatment with DMF as well as three and six months following the treatment M3 and M6 $n = 12$). Volcano plots illustrating differences in DNA methylation between different time points following DMF treatment. Hyper- and hypo-methylated CpGs with min 5% methylation change and $P < 0.001$ (Linear model testing) are indicated in red and blue, respectively. **b** Heat map of 1614 most significant differentially methylated CpG sites between the time points (the scale represents $z$-score). **c** Clusters of canonical pathways, derived using Ingenuity Pathway Analysis suggest functions that are affected in monocytes following six months of DMF treatment. The heat-map depicts cluster 1–3. The significance of pathways was determined with the right-tailed Fisher's exact test ($P < 0.05$) and the clustering was performed on the relative risk for the overlap of molecules between the pathways using Mclust. The horizontal top bar indicates the degree of overlap with selected ROS genes. Summary of the key functions and their constituent genes that displayed changes in DNA methylation upon DMF treatment are given to the right

Non-classical CD14[−]CD16[+] monocytes have been shown to be the subset most vulnerable to ROS-induced apoptosis[55]. Herein we only detected a change in numbers classical inflammatory monocytes (CD14[++]CD16[−]), which were reduced in DMF non-responders (Fig. 2). Since classical CD14[++]CD16[−] monocytes are also positive for CCR2, which is used to recruit myeloid cells into the CNS, it cannot be excluded that differences in cell numbers can be explained by differences in migration rather than

increased apoptosis. DMF-dependent CNS migration of myeloid cells has been described[56] and is further supported by our long-itudinal DNA methylation changes in relevant pathways involved in migration and communication (Fig. 3). However, given the complex interaction between several epigenetic modalities in the regulation of transcription and cellular functions, addressing histone modifications and non-coding RNAs might provide additional insight into DMF-mediated effect at the molecular

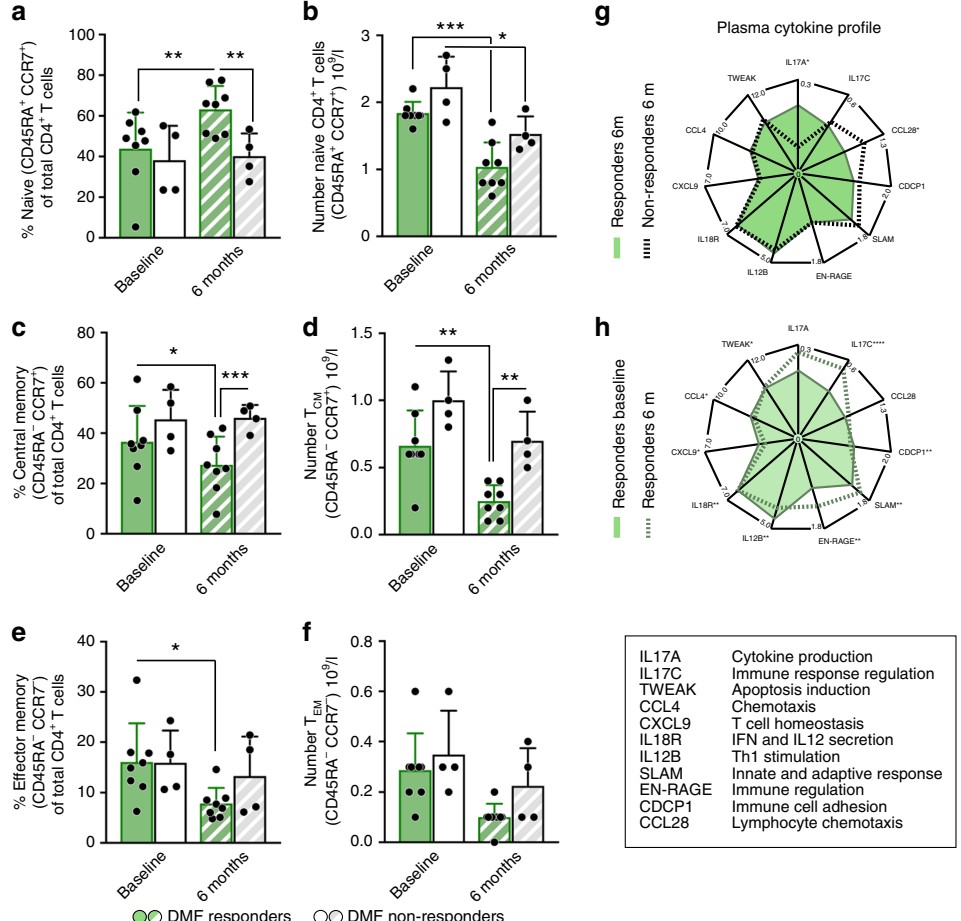

**Fig. 6** DMF responders show altered levels of CD4$^+$ T cells subsets compared to non-responders. **a**, **b** Percentage and absolute number of naïve cells of CD4$^+$ T cells was analyzed in DMF responders ($n = 8$) and non-responders ($n = 4$) at baseline and after six months. **c–f** Central memory T cells (T$_{CM}$) was defined as CD45RA$^-$CCR7$^+$ and effector memory T cells (T$_{EM}$) as CD45RA$^-$CCR7$^+$. Graphs show means and S.D. and all analysis between paired patients are performed with paired $t$ test. **g**, **h** Normalized protein expression (NPX) in plasma of IL17A, IL17C, CCL28, CDCP1, SLAM, EN-RAGE, IL12B, IL18R CXCL9 CCL4 and TWEAK were analyzed in responders ($n = 29$) and non-responders ($n = 9$) at baseline and 6 months after DMF intervention. $*P < 0.05$, $**P < 0.01$, $***P < 0.001$. Asterisk in (**d**) indicate $P$ value adjusted for multiple testing. $**$adj$P < 0.01$, $***$adj$P < 0.0001$

level[57],[58]. The functional evaluation showed that DMF increased the inducible ROS generation more in responders than non-responders, which is interesting in light of the observation that NADPH oxidases have been found in active MS lesions and are believed to contribute to tissue injury[46]. However, it is likely that the consequences of ROS generation are different depending on whether it triggers redox regulation or oxidative damage. Also, the site of ROS generation is likely a crucial factor since the consequences of ROS generation in the CNS parenchyma likely differ compared to secondary lymphoid organs.

Additional evidence for an active role of monocytes in translating the effect of DMF into clinical benefit comes from the genetic association study, which identified a SNP (rs6919626) located in the *NOX3* gene. This SNP was associated both with ROS generation in monocytes and the likelihood of having an adequate treatment response with DMF, in particular since the minor allele was linked with reduced ROS generation and higher risk of breakthrough disease (Fig. 4). *NOX3* does not associate with MS incidence[59] ($P = 0.527$, OR $= 1.01$) and to the best of our knowledge this is the first such association between a functional effect on ROS generation and the clinical response to therapeutic intervention in any autoimmune disease. This observation also lends support to the substantial amount of preclinical studies showing a regulatory role of myeloid derived ROS

on adaptive immunity[22],[27],[28],[31]. Evaluation of the association of rs6919626 with methylation and expression suggested a potential genetically driven influence of the promotor methylation on subsequent transcription of *NOX3* in monocytes. Molecular connection between DMF, *NOX3*, and DNA methylation is relevant, however, also very complex as existing data on the direct influence of fumarates on DNA methylation is still scarce. Conversely, some studies have indicated that both DNA methylation and histone acetylation can influence Nrf2 and its inhibitor Keap1, at least in non-immune cells. Future studies have to verify the role of these changes in an inflammatory context. At this stage a conservative interpretation of our findings in the context of existing knowledge is that DMF increases oxidative functions in monocytes, which are known to modulate T cell functioning, leading to changes in methylation patterns in both cell types. However, our findings need to be verified in additional cohorts in order to fully understand the role of monocytic *NOX3* during DMF intervention.

Our hypothesis of monocytes being primarily affected by DMF is also supported by the temporal profile of methylation changes in a smaller sample set, where changes in CD14$^+$ monocytes occurred prior to changes in CD4$^+$ T cells. Moreover, changes in monocyte numbers occurring prior to changes in lymphocyte numbers in the larger clinical cohort further support monocytes

being targeted by DMF prior to CD4[+] T cells. Myeloid derived ROS has been shown to limit T cell activity[27,28] in vitro. This is a plausible underlying mechanism in our study but cannot confirmed or ruled out herein. However, T cells have recently been described to be under increased oxidative pressure early after DMF intervention[5,17], at a time point which coincides with the peak in monocyte ROS generation found here.

In the CD4[+] T cells we predicted upstream regulators suggesting relevant factors for activation and function of $T_H17$ and $T_{reg}$ (Supplementary Table 5). This was in line with functional implications of wide spread methylation changes validating previous published data on longitudinal changes in $T_H17$ and $T_{reg}$ frequencies in man[18]. Herein IL-6, IL-17A and IL-22, all of which contribute to T-cell subset regulation and MS pathology, were differentially methylated over time in CD4[+] T cells (Fig. 5), all of which contribute to T cell subset regulation and MS pathology.

Changes in the adaptive immune cell profile were further verified with a standardized flow cytometry approach. DMF responders significantly increased their proportion of naïve T cells compared to non-responders. The absolute numbers of naïve T cells did decrease in both groups, suggesting that absolute number of naïve T cells is insufficient to predict beneficial treatment outcome. Lowering of $T_{CM}$ and $T_{EM}$ in RRMS patients with ongoing disease has been described before[18]. Our data further impliy that reduction of $T_{CM}$ could be relevant for beneficial DMF response. The significant difference in $T_{CM}$ numbers between responders and non-responders could further be supported by the differentially methylated genes involved in T cell differentiation, clonal expansion (Hippo) and T cell apoptosis (Fig. 5c). Interestingly, these pathways were also highly influenced by oxidative stress. Pathways involved in lymphocyte trafficking and migration are also changed over time and between responders and non-responders (Figs. 5c, 6g). These pathways could be of importance in e.g., $T_H17$ and $T_{reg}$ migrating to the CNS, leaving the blood. Since also $T_{CM}$ and $T_{EM}$ in the CNS are associated with disease progression, and are decreasing in blood of responders, CNS migratory pathways are likely not of relevance in $T_{CM}$ and $T_{EM}$ in our cohort. Decreased migration would more likely cause an accumulation in the blood[18].

In conclusion, we here demonstrate that monocytes undergo functional, numeric and DNA methylation changes early after initiation of DMF. This is, at least in part, related to their oxidative capacity and occur prior to immunomodulatory changes to pathways in CD4[+] T cells associated with MS pathogenic mechanisms. A direct link between DMF-mediated modulation of monocytes and the clinical effect of DMF is suggested both by an association between the early monocyte phenotype and clinical outcomes, as well as by the identification of a genetic locus influencing both oxidative capacity in monocytes and clinical outcome of DMF treatment. Although the action of DMF on the exceedingly complex regulation of redox state is likely to involve both anti-oxidant and oxidant effects, the fact that patients starting DMF display increased levels of non-enzymatically produced isoprostanes strongly supports a net increase in oxidative state, at least in the peripheral blood. Collectively, these findings challenge the widespread belief on oxidants and anti-oxidants as categorically detrimental or beneficial in conditions of autoimmunity. On the contrary, this discovery may pave the way for work aiming to modulate and increase ROS generation in monocytes as a therapeutic strategy to control dysregulated autoimmune responses.

## Methods

**Study population.** In total, 564 patients and healthy subjects were sampled during May 2014 and March 2017. All patients had indication for start of DMF in clinical routine and fulfilled diagnostic criteria of RRMS according to the 2010 revision of the McDonald criteria[60], but otherwise were not subject to any other selection criteria. Patients were either categorized as DMF responders if they had continuous DMF therapy for at least 24 months without signs of disease activity as defined by being free of clinical relapses and newly appearing cerebral MRI lesions, or DMF non-responders if they displayed signs of continued clinical and/or neuroradiological disease activity at any stage after the first three months. Patients having disease activity within three months after starting DMF or lacking a baseline sample before DMF start were excluded from the study. Baseline characteristics of the cohorts, analysis inclusion and ethical permissions are summarized in Supplementary Dataset 1. This study was a part of the Stockholm Prospective Assessment of MS study (STOPMS II) (2009/2107–31/2) and IMSE (2011/641-31/4), approved by the Regional Ethical Review Board of Stockholm, all participants provided written consent.

**Flow cytometric analyses and intracellular ROS generation.** Peripheral blood was sampled in EDTA tubes and analyzed within an hour of sampling. Monocyte subsets were analyzed in responders ($n = 10$) and non-responders ($n = 10$). Erythrocytes were lysed using Isolyse C (Beckman Coulter, Brea, CA) and stained with CD14-PE.Cy7 (A22331) and CD14-FITC (IM0814U) (Beckman Coulter, Brea, CA) for 30 min at +4 °C in dark. For examination of intracellular ROS generation, samples were prepared and analyzed with Phagoburst kit (BD Bioscience, Franklin Lakes, NJ) according to the manufacturer's standard protocol. In brief, whole blood from healthy donors ($n = 10$), RRMS patients ($n = 20$) before and after DMF therapy was either ex vivo stimulated with dilutions of E. coli or PBS before measuring intracellular ROS generation with dihydrorhodamine-123 (DHR-123). Supplementary Dataset 1 defines patients included for genetic association with ROS generation. All samples were analyzed with a 3 laser Beckman Coulter Gallios using Kaluza Software (Beckman Coulter, Brea, CA) and acquired by time. Numbers of monocytes and lymphocytes in Fig. 2e, f were determined by differential blood count performed according to clinical routine at the Karolinska University Hospital Laboratory. Characterization of naive CD4[+] T cells (CD45RA[+]CCR7[+]), $T_{CM}$ (CD45RA[−]CCR7[+]) and $T_{EM}$ (CD45RA[−]CCR7[−]) subsets were performed at Clinical Immunology/Transfusion medicine, Karolinska University Hospital Laboratory, Huddinge, Sweden using a FITMaN/HIPC-based standardized phenotyping panel developed by the Human Immunophenotyping Consortium (HIPC)[61]. In brief, frozen PBMC responders ($n = 8$) and non-responders ($n = 4$) were stained with CD3-V450 (UCHT1), CD4-PerCP-Cy5.5 (RPA[−]TA), CD8-APC-H7 (SK1), CD45RA[−]PE-Cy7 (HI100), CD45-AF700 (HI30) and CD197/CCR7-PE (150503) (BD Bioscience, Franklin Lakes, NJ) and analyzed with a 3 laser Beckman Coulter and Kaluza Software (Beckman Coulter, Brea, CA).

**8.12-iPF2α-VI isoprostane and cytokine analysis in plasma.** The plasma levels of free 8.12-iPF2α-VI isoprostane were extracted and quantified in plasma as previously published with minor modifications[62,63]. The extraction of free 8.12-iPF2α-VI was performed on an Extrahera automated extraction system from Biotage (Uppsala, Sweden) as previously described with minor modifications. Briefly, 10 μl of an antioxidant solution mix [0.2 mg/mL BHT and EDTA in MeOH:Water (1:1)] and 10 μl of the internal standard solution were added to a 12 × 75 mm Pyrex tubes on kept on ice. Then, 400 μl of plasma thawed at 4 °C was added to the tubes and samples were vortexed. Afterwards, 600 μl of the extraction buffer (0.2 M $Na_2HPO_4$:0.1 M citric acid, in water, pH 5.6) was added and samples were vortexed. Samples were immediately extracted using 3cc/60 mg HLB Oasis SPE cartridges (Milford, MA) previously conditioned with 2 mL of methanol followed by 2 mL of water. Samples were then loaded into the cartridges, washed with 3 mL of water/methanol 90:10 and eluted with 2.5 mL of methanol. The solvent was then evaporated using a $N_2$ Turbovap LV system (Biotage, Uppsala, Sweden). Samples were finally reconstituted in 80 μl of methanol:water (50:50, v/v), filtered using Amicon Ultrafree-MC, PVDF 0.1 μm filters (Millipore, US) centrifuged at 4000 × g for 5 min and transferred to LC-MS vials with 150 μl inserts for analysis. The isoprostane analyses were performed on an ACQUITY UPLC System from Waters Corporation (Milford, MA) coupled to a Waters Xevo® TQ-S triple quadrupole system equipped with an electrospray ion source operating in the negative mode. Separation was adapted from a previously published method and carried out on a Zorbax RRHD Eclipse Plus C18 (100 × 2.1 mm, 1.8 μm, 100 Å) column equipped with a guard column (5 × 2 mm), both from Agilent Technologies (Santa Clara, CA). Mobile phases consisted of 0.01% acetic acid in water (aqueous) and 0.01% acetic acid in methanol (organic). The elution gradient used was as follows: 0.0 min, 40.0% B; time range 0.0 to 8.9 mins, 50.0 → 58.5% B; time range 8.9–9.1 mins, 58.5% B; time range 9.1–9.6 mins, 58.5 → 100% B; time range 9.6 to 11.5 min, 100% B; time range 11.5–13.3 min,100 → 50% B and 13.2. The flow was maintained at 50% B until 14 min. The flow rate was set at 250 μl min[−1], the injection volume was 7.5 μL and the column oven was maintained at 35 °C. SRM transitions 353 → 115 and 357 → 115 were monitored for 8,12-iso-iPF2α-VI and 8,12-iso-iPF2α-VI-d$_4$ obtained from Cayman Chemicals (Ann Arbor, MI) and used for their quantification. Samples were extracted in three different batches and injected in one LC-MS/MS batch. In order to minimize sample manipulation effects on the quantification, paired samples were extracted and injected consecutively in a randomized order (Baseline-6 months/6 months-Baseline). In order to control for reproducibility within the same batch, pooled plasma QC mixtures were prepared using 60 μl for each sample (whenever enough volume was left). A

triplicate of the QC was extracted and injected for each batch. The %CV of 8,12-iso-iPF2α-VI for each batch were 5.6, 10.2 and 11.2%, so the data can be considered as reproducible. For cytokine analysis, samples were analyzed with the Immune Response Panel by Olink Proteomics, Uppsala, Sweden using the proximity extension assay for quantifying relative cytokine levels.

**Cell sorting and transcriptomics analysis.** CD4+ and CD14+ cells where prepared within an hour after sampling using AutoMACS (Milteny Biotec, Bergisch, Germany) according to the manufactures standard protocol and stored at −70 °C. DNA and RNA was extracted simultaneously using Qiagen Allprep DNA/RNA kit (Qiagen, Venlo, Netherlands) according to the manufactures standard protocol and stored at −70 °C. This kit does not allow extraction of short RNAs. RNA was analyzed on GeneChip Human Gene 2.1 ST Array from Affymetrix by NGI in Uppsala, Sweden. DNA methylation was measured on the Illumina EPIC array by BEA in Stockholm. Affymetrix data was loaded using the oligo and affy R-packages. Data was RMA normalized and batch effects were determined using PCA. Differential expression was determined using the same linear model used for methylation. GSEA was performed using GenePattern (https://genepattern.broadinstitute.org) and GO_REGULATION_OF_RESPONSE_ TO_OX-IDATIVE_STRESS on expression in paired CD14+ monocytes before and after DMF ($n = 7$). CD14+ Affymetrix data was loaded using the oligo and affy R-packages. Data was RMA normalized and batch effects were determined using PCA. Differential expression was determined using the same linear model used for methylation.

**DNA methylation analysis.** DNA methylation data was loaded using the ChAMP package[64] https://github.com/TranslationalBioinformaticsUnit/GeneSetCluster. Data were processed as previously described[65], in brief, data was loaded with all probes passing the detection $P = <0.01$ (Linear model testing), furthermore all probes with known SNPs were removed. Notably because the study design is before and after treatment, the X and Y chromosome probes were not filtered. After this the betas were normalized using BMIQ, and afterwards the batch effects were identified with PCA and removed using combat[66]. Differential methylation was determined using a paired linear model, using the limma package[67]. The linear model used age, sex and cell type deconvolution as covariates. Cell type deconvolution was performed using RefFreeEWAS R-package. The optimal number of cell types for deconvolution was determined by calculating the deviance-boots (epsilon value) over 10000 iterations for the range of 1 to 10 cell types (https://CRAN.R-project.org/ package = RefFreeEWAS). The cell proportions are obtained by solving the model $Y = M*\Omega^{-T}$ (where $Y$ = original beta methylation matrix, $M$ = cell type specific beta methylation matrix, $\Omega$ = cell proportion matrix and $T$ is number of cell types to deconvolute) using nonnegative matrix factorization method[68].

**Pathway analysis.** Differentially methylated genes were uploaded to Ingenuity Pathway Analysis (IPA). Core expression was performed and canonical pathways were grouped into clusters by calculating the similarity of pathways by calculating the relative risk (RR) of each pathway appearing with each pathway based on the molecules within the pathway. RR scores were clustered into groups using MClust. ROS-score per clusters was calculated by calculating the number of genes that match with the genes from GO_RESPONSE_TO_OXIDATIVE_STRESS. Differentially expressed genes were also uploaded to IPA, including the direction of change in the analysis.

**Genetic association study.** Genotyping was carried out using an Illumina custom array as part of a larger study replicating genetic association to MS within the international multiple sclerosis genetics consortium (IMSGC) (https://www.biorxiv.org/content/early/2017/07/13/143933). A total of 7701 multiple sclerosis patients and 6637 controls from Sweden were genotyped and passed quality control. Allele calling was carried out with an Illuminus caller. The quality control analyses for markers included minor allele frequency (MAF) > 0.02, success rate > 0.98, Hardy–Weinberg equilibrium among controls ($P = <0.0001$). For individuals, the quality control included success rate > 0.98, increased heterozygosity as determined as F (inbred coefficient) smaller than mean value minus three standard deviations. All these quality control steps were carried out using PLINK. We identified population outliers using the SmartPCA program with standard settings and removed those that were outliers. Eleven PCA vectors, those with $P = <0.05$, were used for correcting for population stratification in the association analysis. We estimated relatedness between individuals using PLINK and removed one individual in reach pair with Pi_hat > 0.175.

Genes encoding some of the components of the NADPH oxidase 1–4 complexes had been tagged using HaploView and single nucleotide markers (SNPs) added to the custom genotyping array (Supplementary Dataset 4), these were included in the association analysis. Analysis for association (response to DMF) were carried out with logistic regression in PLINK v1.07 including eleven PCA vectors to correct for population stratification. 323 markers and 341 subjects were included after QC (Supplementary Dataset 1).

Measurements of ROS production was performed as described in methods section using the Phagoburst Kit (BD Bioscience, Franklin Lakes, NJ). 114 subjects were included (Supplementary Dataset 1), and genotypes were available for 204 of

the selected SNPs. A quantitative trait analysis was performed in PLINK, including eleven PCA vectors to correct for population stratification.

The Linkage Disequilibrium plot of the markers in *NOX3* was made using HaploView 4.2 using genotypes from 7701 MS patients and 6637 controls from the Swedish population.

**Statistics.** General statistical analysis was performed in GraphPad Prism software. Throughout the study *n* refers to the number of subject where every subject is one data point. Two group comparisons were done with Student's two-tailed unpaired *t* test. Paired group comparisons were done with paired *t* test. Two group comparisons with a control group were done with one-way ANOVA. $P < 0.05$ was throughout considered statistically significant. Additional statistical methods applied for genetic association and transcriptional and epigenetic characterization are described in each individual section. Group and cohort sizes are indicated in figure legends and Supplementary Dataset 1. Violin plots were generated using R software and Plotly package.

**Reporting Summary.** Further information on research design is available in the Nature Research Reporting Summary linked to this article.

## Data availability

Source data on methylation and transcription used to generate graphs in Figs. 1, 3, 4, 5 have been deposited in GEO database under accession number GSE130494 (methylation data) and GSE130478 (transcription data) Genetic data is stated to consist of personal data in the GDPR law. The data protection officer at Karolinska Institute interpretation is that genetic data if from more than 30 or so polymorphisms, could identify a person and hence cannot be anonymized. Thus we herein provide genetic data from the two highest associated SNPs in NOX3 in Supplementary Dataset 4, 5 and upon request we agree to share additional data. The human genotypes herein is part of a larger MSchip study (https://www.biorxiv.org/content/early/2017/07/13/143933), no authors from that study claims co-authorship to this study.

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

## Acknowledgements

This study received funding from the Swedish Medical Research Council (grant 2017–03054), Stockholm County (grant 20170216), by an unrestricted academic research grant from Biogen, Hjärnfonden, NEURO Förbundet and Karolinska Thematic Center in Inflammation.

## Author contributions

K.E.C., T.O., M.J., F.P.: conceived the study and K.E.C., F.P. wrote the manuscript with support from co-authors. K.E.C. summarized data and performed DNA/RNA extraction from sorted cells, expression analysis and performed flow cytometry with support from S.A. S.A. performed cell sorting. S.L.E. performed and analyzed FITMaN characterization. E.E., D.G-C., M.J. performed methylation and expression analysis, with support from T.V.S.B., M. Gustafsson. M. Granqvist, F.A. prepared plasma and PBMC samples for further analysis and collected and summarized patient descriptive data and cell counts. A.G., I.K. performed genetic association studies. J.H. analyzed cytokines. AC/CW measured isoprostanes. The funders were not involved in study design, data interpretation or manuscript writing.

## Additional information

**Competing interests:** F.P. has received research grants from Biogen, Novartis, and Genzyme, and fees for serving as Chair of DMC in clinical trials with Parexel. T.O. has received unrestricted MS research grants, and/or lecture advisory board honoraria from Biogen, Novartis, Sanofi and Roche. The remaining authors declare no competing interests.

**Peer review information**: *Nature Communication* would like to thank the anonymous reviewers for their contributions to the peer review of this article. Peer review reports are available.

