## [Peer Review File · Nature Communications]

Reviewers' comments:

Reviewer #1, expertise in human monocytes (Remarks to the Authors):

The manuscript provided by Carlström et al. describes the effect of DMF on monocytes and links its therapeutic mechanism to increased ROS production.

This is an interesting manuscript which needs further improvement before publication can be recommended.

1. The different observations are poorly linked: In a manuscript published in a journal like Nature Communications, I would expect more mechanistic data. What is the molecular connection between DMF, NOX3, and DNA methylation?
2. To prove that the NOX3 SNP is indeed the reason for non responding to DMF treatment many controls are missing, e.g. experiments with mutated cells harbouring this SNP. Moreover, NOX3 is not exclusively expressed in monocytes; what happens in other cell types? Furthermore, what is the phenotype of the SNP, does it show any connection to MS?
3. Today, the usage of the term ROS is a bit problematic since many ROS are known and each of it has different function and activity. Please specify the ROS in introduction and discussion and mention which ROS was measured in your experiments.
4. Sometimes novelty is a bit overrated:
 - since 1974 it is known that hydrogen peroxide acts as second messenger, the latest definition of oxidative stress distinguishes between oxidative eustress (redox regulation) and oxidative distress (oxidative damage), so the classification of oxidants and antioxidants in detrimental and beneficial is already disproved
 - the idea that DMF acts as an oxidant has been stated before as well
 - increasing ROS is a well established therapeutic tool in many diseases

Reviewer #2, expertise in clinical multiple sclerosis (Remarks to the Author):

Carlström et al. examine the mechanism/s of action of dimethyl fumarate (DMF) in multiple sclerosis (MS) patients treated with this drug. They focus on the effects of DMF on the transcription factor Nrf2 and how it may influence the production of reactive oxygen species (ROS) in different immune cells. Their surprising findings are an increase of ROS in monocytes despite that known or assumed anti-oxidative effects of DMF and Nrf2, alterations in numbers, expression and epigenetic profiles of monocytes in monocytes prior to T cells, and finally a single nucleotide polymorphism (SNP) in the ROS-producing NOX3 gene.

The findings are interesting, but also preliminary. The authors develop the study based on the assumption that the DMF primarily acts via its influence on Nrf2 and the effects of this transcription factor on the redox system. When they describe the known mechanisms of DMF it is surprising that they do not mention a seminal study by Ghoreschi et al. (Fumarates improve psoriasis and multiple sclerosis by inducing type II dendritic cells, *J. Exp. Med.* 2011). This study, which appeared seven years ago, dissected the effects of fumarate on dendritic cell- (DC) and T cell differentiation in detail, describes glutathione (GSH) depletion, induction of type II DCs, effects on hemoxygenase I, increased Th2 differentiation and others. Since this work has not been mentioned and considered, the focus of the study of Carlström et al. remains narrow.

The study builds on blood samples collected from a large number of DMF-treated MS patients, which is a strength of the study. Plasma levels of free isoprostane are used as a measure of oxidative state and stress, and interestingly the levels increase rather than decrease during DMF therapy. This finding is supported by gene expression changes in CD14+ monocytes, which demonstrate an upregulation of genes involved in oxidative stress. Here, the differential gene expression data could have been described in more detail and not only in an overview/summary fashion. When reading the text and figure, it is not clear what has been done to examine ROS production *ex vivo*, until one goes through

the material and methods section and finds that DHR stands for dihydro-rhodamine. The intervals, at which blood was collected, are not clear from the first paragraph of the results. Also, it would be interesting to know if there are only full responders and vice versa non-responders or also patients with an incomplete response as is well known for the beta interferons, and how many patients fall into such a category if it exists. The changes in monocytes precede alterations in T lymphocytes, and therefore the authors assume that one depends on the other, but it is not clear if these associations are due to functional interactions of the two cell types. The changes in T cell differentiation states are described in a global fashion, but it remains unclear, how what causes the relative increase in naive cells in responders. Fig 5 shows that the percentages of central- and effector memory cells differ between responders and non-responders, but this is not related to absolute lymphocyte counts, which drop substantially in DMF-treated patients. Fig. 5 C is poorly explained, and it is not clear, what the absolute values of cytokines are that have been measured. If the values around the graph and for each cytokine were plasma levels, e.g. 0.3 pg/ml for IL-17A, these values would be very low and question the biological relevance of the changes.

Sentences like: "Functional implication of wide spread methylation changes detected in our study further validate previous data in man on longitudinal changes in Th17 and Treg frequencies (24), which have major impact on MS pathology.", are too far-reaching/not supported by the literature on disease-relevant T cells in MS and therefore should be omitted or more specific. As already mentioned, it is difficult to judge changes of naive and memory cell populations unless the absolute lymphocyte counts are considered.

For many of the measures, the numbers of samples are relatively, sometimes very small, which surprises when considering the large number of patients that have regularly been sampled. One therefore wonders how representative the observations are for the entire cohort of patients.

With respect to the NOX3-associated SNPs and their relation to the findings, it would be good to give the reader a sense for the number of individuals out of the entire DMF-treated cohort, which had been tested (either in the text of the figure).

Reviewer #3, expertise in ROS and epigenetics of autoimmune diseases (Remarks to the Author):

COMMENTS TO AUTHORS

The Authors sought to investigate the mechanism of Dimethyl fumarate (DMF) on circulating monocytes in patients with RRMS. They have shown that monocytes increase their ROS production after DMF intervention. Moreover, monocytes changed in numbers, expression and methylome profiles, prior to changes in T cells. Finally, they identified a possible single nucleotide polymorphism in the monocytic ROS-producing NOX3 gene associated with DMF treatment response and ROS production. Then, they proposed the mechanisms of DMF and monocytes in autoimmune diseases.

Major Comments:

1) Overall, the paper of Carlstrom et al. is quite interesting, but the level of presentation is inaccurate and somewhat poor (for example, the list of references is very confusing as well as the definition and pixels of Figure 1 is unacceptable for a high-level reputation Journal). The description of oxidative stress occurring in the CNS is poor. In fact, the content of oxygen radical scavengers is higher in the brain when compared to other arterial districts (see and quote, Napoli C et al. *Circulation* 1999; D'armiento FP et al. *Stroke* 2001). In general, ROS-sensitive mechanisms are balanced by endogenous oxygen radical scavengers in the brain (Sun N et al. *Mol Cell* 2016; Napoli and Palinski, *Neurobiol Aging* 2005) and this issue appears also to be relevant in the pharmacological action of DMF. Thus, in the setting of isolated monocytes from patients treated with DMF, additional experiments should be carried to better understand the effect of ROS in the presence of specific scavengers and/or substances inhibitors of ROS.

2) The measure of global epigenetic changes is not well represented by DNA methylation alone (in fact, data on histone acetylation as well as miRNA and non coding RNA should be included). Otherwise, you need to change the term "global epigenetic changes" to "DNA methylome" throughout

the paper. Also in this case, there is no appropriate mention in the the Introduction of why the Authors retained relevant to study such epigenetic-related events. Thus, the Authors need to mention relevant epigenetic studies in the field (see and quote: He H et al, Hum Genomics 2018; Wang Z, et al. Curr Opin Rheumatol. 2017; Picascia A et al. J Neuroimmunol 2015) and explain why you chosen to focus your attention on DNA methylation alone. Finally, the issue of epigenetics in autoimmune disease is even more complicated (see and quote: Wu H, et al. Autoimmun Rev. 2018; Ray and Yung. Clin Immunol. 2018; Shu Y, et al. Clin Rev Allergy Immunol. 2017; Picascia A et al. Clin Immunol 2015) and, consequently, the pathogenic effects of ROS should be dissected in epigenetic and non epigenetic-sensitive mechanisms. Also in this setting, additional experiments should be carried, in order to undestand which specific mechanisms are ROS-mediated and which are subjected to epigenetic drive. The general statement that monocytes are involved in autoimmune diseases should be redirected to the findings of the present study (only in the Discussion).

SPECIFIC COMMENTS

- 1) In the title page, it was mentioned that none of the Authors had conflicts of interest. However, in Notes, it was declared that FP received grants from Biogen, Novartis, and Genzyme and fees for serving as Chairs in clinical trials with Parexel.
- 2) ABSTRACT: The abstract is poorly written. Major molecular mechanisms should be included.
- 3) Figure 1 should be replaced by 1200dpi acquired images.
- 4) Delete Panel B of Figure 2.
- 5) The list of References must be integrated and renumbered.

Reviewer #1

-Not clear how ROS in monocytes have been assessed and at what time-points

Action: The use of DHR123 to measure intracellular ROS has been clarified in the results and methods sections. In addition, to help the reader, a graphical overview of time points at which samples for different tests have been collected has been added as Fig1 .A and B.

Page 5 Results

“Next, we determined monocytic ROS generation in RRMS patients and controls using dihydrorhodamine-123 (DHR-123). Baseline levels of spontaneous ROS generation before starting DMF, although very low, were higher in RRMS patients than healthy controls, and spontaneous ROS generation further decreased after starting DMF independent of later being a responder or non-responder (Fig.2H)”

Fig.2 legend

“Histogram shows representative DHR-123 MFI \pm S.E.M. at three months (A)”

“Spontaneous generation of ROS in healthy controls (n=10) and DMF untreated (n=18) and treated (n=18) patients (H), and ROS generation in ex vivo-stimulated monocytes from DMF responders (n=10) and non-responders (n=7) measured with DHR-123 (I).”

Reviewer #1

“*What is the molecular connection between DMF, NOX3, and DNA methylation?*”

Action: We here included the genome-wide DNA methylation (and mRNA expression) analyses in monocytes and CD4+ T cells as a functional genomic readout of a degree of modulation in these two relevant cell types. Similarly, Maltby et al 2018 published a study on DNA methylation in CD4+ T cells in a smaller cohort of RRMS patients, with sampling at baseline and at six months after starting therapy. Our findings here are largely in agreement with those of Maltby et al, with the important addition that changes in CD4+ T cell methylation patterns become prominent only after the first three months, a time point at which the clinical effect of DMF also becomes evident. We agree with the reviewer that the question regarding the molecular connection between DMF, *NOX3*, and DNA methylation is relevant, however, also very complex as existing data on the direct influence of fumarates on DNA methylation is still scarce. Conversely, there are some studies that have indicated that both DNA methylation and histone acetylation can influence Nrf2 and its inhibitor Keap1, at least in non-immune cells. At this stage a conservative interpretation of our findings in the context of existing knowledge is that DMF increases oxidative functions in monocytes, which are known to modulate T cell functioning, leading to changes in methylation patterns in both cell types. These methylation changes might impact the transcription, and thus directly the functional activity, and/or be a consequence of the transcriptional activity due to the changed cellular functioning. Although we cannot infer the causality of detected changes under this set-up, the identified DNA methylation changes are still very useful in uncovering the type of functional changes and pathways that occur in these cells during treatment with DMF. A novel finding here is the genetic association of oxidative capacity in monocytes as well as clinical efficacy with *NOX3*. To provide a full understanding of how oxidative reactions, modulated by DMF, and in part executed by *NOX3* (see also below) lead to changes in methylation patterns, which in turn may provide a feed-back regulation, is outside of the scope of this study.

Reviewer #1

“To prove that the *NOX3* SNP is indeed the reason for non-responding to DMF treatment many controls are missing, e.g. experiments with mutated cells harbouring this SNP.”

Action: We here provide suggestive evidence for an association between a SNP in *NOX3* and oxidative capacity in monocytes as well as clinical efficacy. The connection to clinical efficacy is virtually impossible to address with experimental models (both *in vitro* and *in vivo*), since the SNP connects a genetic variation with a very complex *in vivo* outcome in humans. Hence, the International MS Genetics Consortium (IMSGC), in which we take part, has identified >200 such variations associated with disease risk, but examples providing formal experimental proof for how a disease-associated human SNP mediates disease risk are limited to a few genes. Instead, such proof builds on integration of data from different studies using multiple approaches. Still, a genetic association provides the best evidence of causality according to current praxis and a first step towards providing more firm proof is to give other groups a chance to replicate our data in independent cohorts. We have now added this to the discussion.

In addition, we have elaborated on the mechanisms downstream of genetic variations in rs6919626 to provide a link between *NOX3* and response to DMF. Albeit limited sample size, we have investigated associations of the SNP in *NOX3* with the levels of methylation in the gene regulatory regions of the *NOX3* gene and *NOX3* expression in our cohort. The new findings describe that the allele associating with a high ROS production and a lack of response to DMF shows evidence of association with decreased methylation levels at several CpGs in the promoter region of *NOX3* and a tendency of higher *NOX3* expression in monocytes, in line with the canonical role of DNA methylation in regulating gene expression, and in our mind addresses a major part of the Reviewer’s concern.

Page 7 Results

“Thus in order to suggest a mechanistic rationale for genetic variations in rs6919626 in association to ROS and therapy outcome we assessed methylation in the *NOX3* promoter region and expression of *NOX3* in sorted CD14⁺ monocytes. The A allele that associated with the response to DMF demonstrated consistent tendency for association with lower methylation at several CpGs in the promoter of *NOX3* (Fig.4C), already at baseline (Fig.4B). At 3 months we also recorded a peak in ROS generation (Fig.2I), which is in line with previous data describing transcription of anti-oxidative response at this particular time-point²⁵. The suggestively lower degree of CpG methylation in *responders* with the A allele at rs6919626 was further reflected in higher *NOX3* transcription in CD14⁺ monocytes at 6 months (Fig.4D). Together, this suggests that genetic variation and CpG methylation in monocytic *NOX3* might influence the *NOX3* gene transcription and thus monocyte function.”

Page 10 Discussion

“*NOX3* does not associate with MS incidence⁶¹ ($P=0.527$, OR=1.01) and to the best of our knowledge this is the first such association between a functional effect on ROS generation and the clinical response to therapeutic intervention in any autoimmune disease. This observation also lends support to the substantial amount of pre-clinical studies showing a regulatory role of myeloid derived ROS on adaptive immunity^{30,35,36,39}. Evaluation of the association of rs6919626 with methylation and expression suggested a potential genetically driven influence of the promoter methylation on subsequent transcription of *NOX3* in monocytes. However, our findings need to be verified in additional cohorts in order to fully understand the role of monocytic *NOX3* during DMF intervention.”

Reviewer #1

“*NOX3* is not exclusively expressed in monocytes; what happens in other cell types? Furthermore, what is the phenotype of the SNP, does it show any connection to MS?”

Action: These are relevant points, and they also relate to the previous question. Regarding disease risk, we have now checked this association using the most recent MS case-control data set, since previous analyses of chromosome X have had flaws. *NOX3* does not associate to MS-risk ($P=0.527$, $OR=1.01$). This study includes 14 802 cases and 26 703 controls from 15 different studies (<https://www.biorxiv.org/content/10.1101/143933v1>).

Regarding the role of the *NOX3* SNP in other cell types we have now added a brief description of corresponding results in granulocyte. The SNP rs6919626 is not associated to ROS generation in granulocytes and other SNPs with nominal association to ROS generation in granulocytes are not associated to clinical response. This does not exclude that part of the effect of DMF can be mediated by other cell types, however, it is of interest that in prior work mostly monocytes and myeloid derived cells have been connected with modulation of T cell functioning in autoimmune disease models. In addition, granulocytes and lymphocytes seem to have a fairly constant ROS generation prior and following DMF (Fig.S1). The observation that the rs6919626 SNP is associated with the methylation pattern of the *NOX3* promoter region in monocytes also strengthens this notion.

Page 7 Results

“Thus in order to suggest a mechanistic rationale for genetic variations in rs6919626 in association to ROS and therapy outcome we assessed methylation in the *NOX3* promoter region and expression of *NOX3* in sorted CD14⁺ monocytes. The A allele that associated with the response to DMF demonstrated consistent tendency for association with lower methylation at several CpGs in the promoter of *NOX3* (Fig.4C), already at baseline (Fig.4B). At 3 months we also recorded a peak in ROS generation (Fig.2I), which is in line with previous data describing transcription of anti-oxidative response at this particular time-point²⁵. The suggestively lower degree of CpG methylation in *responders* with the A allele at rs6919626 was further reflected in higher *NOX3* transcription in CD14⁺ monocytes at 6 months (Fig.4D). Together, this suggests that genetic variation and CpG methylation in monocytic *NOX3* might influence the *NOX3* gene transcription and thus monocyte function.”

Reviewer #1

“the latest definition of oxidative stress distinguishes between oxidative eustress (redox regulation) and oxidative distress (oxidative damage), so the classification of oxidants and antioxidants in detrimental and beneficial is already disproved

- the idea that DMF acts as an oxidant has been stated before as well
- increasing ROS is a well-established therapeutic tool in many diseases”

Action: We thank the reviewer for clarifying these issues. We have updated the text accordingly. Regarding the second and third statement the authors agree with the reviewer and we have clarified the outcome on the oxidative environment from DMF (see below). Regarding the beneficial role of ROS we now cite several of these key publications.

Page 3 Introduction

“Engagement of responses to redox and oxidative damage within the human CNS is also crucial and likely multifactorial. As an example, anti-oxidative proteins are found in active

MS lesions^{52,53}. However, the aim of this particular study was to perform a detailed characterization of the initial monocyte response and subsequent immunomodulation occurring in peripheral blood of RRMS patients starting therapy with DMF.”

Page 2 Introduction

“The net activity of DMF has been described to be mainly anti-oxidative^{8,13,14}. However, since the Nrf2 is a sensor for oxidative stress the activation of Nrf2 most likely involves transient oxidative stress by DMF. There are also discrepancies in the literature how to assess oxidative stress and whether fluctuations of anti-oxidative transcripts or proteins reflect decreased or increased oxidative stress^{15,16}.”

Reviewer #2

-the authors do not mention the study by Ghoreschi *et al.* 2011 describing role of fumarate on DC and T cells but also describes GSH depletion

Action: We regret this reference was omitted in the original submission, since their findings regarding GSH depletion, a classical way of quantifying oxidative stress, are a relevant correlate to the observations we have here, including for example non-enzymatic oxidation of isoprostanes. This increasingly is seen as a more stable outcome to measure oxidative stress and our results can be seen as a validation of the study by Ghoreshi *et al.* (now acknowledged in the introduction and discussion).

Page 3 Introduction

“DMF has been ascribed cyto-protective effects of potential relevance for CNS cells during inflammation, but conclusive data on degree of CNS penetration in humans is still lacking^{20,21} and modulation of disease relevant T cell subsets²²⁻²⁷, therefore remains the most likely mechanism for reducing clinical and neuroradiological disease activity in RRMS^{8,28,29}”

Page 9 Discussion

“Previous studies have indicated similar changes based on less reliable indirect measurements, such as increased transcription or anti-oxidative protein depletion^{8,25,27}.”

Reviewer #2

-GSEA could be described in more detail.

Action: Thank you for pointing this out. We have now added a heat map of the genes in the gene list, which makes it easier for the reader to understand variation, clustering and trends. In addition, we have given examples of key transcripts in the text.

Page 4 Results

“This was also in line with Gene Set Enrichment Analysis (GSEA) on differentially expressed mRNAs in CD14⁺ monocytes at baseline and after six months, which showed an enrichment of upregulated genes involved in response to oxidative stress as compared to baseline (including *TXN*, *SOD1/2*, *NFE2L2*) (Fig.1E, F).”

Reviewer #2

-Not clear how ROS in monocytes have been assessed and at what time-points

Action: For details, please see response to reviewer #1 above. Briefly, the use of DHR123 to measure intracellular ROS has been clarified in the results and methods sections. In addition,

to help the reader, a graphical overview of time points at which samples for different tests have been collected has been added as Fig1.A and B.

Reviewer #2

“Also, it would be interesting to know if there are only full responders and vice versa non-responders or also patients with an incomplete response as is well known for the beta interferons, and how many patients fall into such a category if it exists.”

Action: We thank the reviewer for pointing this out. Indeed, in RCTs and larger real-world studies comparing different DMTs and/or placebo it is common practice to report degree of therapeutic response (primarily relapses and MRI activity) in relation to run-in disease activity rather than using a dichotomization into responder/non-responder. We here used a relatively small core cohort to investigate laboratory phenotypes in depth and a larger real-world cohort to replicate findings regarding cell counts and to provide a material for the genetic association to clinical outcomes. In both cases, due to the inherent statistical properties of models used to investigate the respective outcomes, the power would be too low for exploring more complex dynamic outcomes. Still, we believe that using a definition of appearance of clinical relapses and/or new MRI activity for an inadequate treatment response is non-controversial, and also adheres to clinical practices (now better clarified in the methods section). In addition, we have added a schematic illustration to Fig.1B relating to the definition of responder/non-responder and also repeated the definition in result section.

Page 5 Results

“Depending on the treatment outcome, patients were either categorized as DMF responders if they had continuous DMF therapy for at least 24 months without signs of disease activity, or DMF non-responders if they displayed signs of continued clinical and/or neuroradiological disease activity at any stage after the first three months (Fig.1B).”

Page 12 Methods

“Patients were either categorized as DMF responders if they had continuous DMF therapy for at least 24months without signs of disease activity, or DMF non-responders if they displayed signs of continued clinical and/or neuroradiological disease activity at any stage after the first three months. Disease activity was herein defined as relapse and/or new MRI lesions. Patients having disease activity within three months after DMF insertion or was not sampled at baseline was excluded from the study.”

Reviewer #2

“The changes in monocytes precede alterations in T lymphocytes, and therefore the authors assume that one depends on the other, but it is not clear if these associations are due to functional interactions of the two cell types.”

Action: The reviewer correctly states that we do not present formal proof for a functional interaction between monocytes and T cells (and that this is a mechanism of action for DMF therapy in RRMS). However, as cited in the manuscript, there are several studies using experimental approaches strongly supporting that ROS generation from monocytes regulates T cell function (*PubMedID:10799876, 20861446, 21593419, 18771940, 20881184, 20682913, 20682913, 18383034, 15310853*). In the discussion we emphasize that our study does not confirm an interaction, but that findings are in line with published experimental studies and thus provide a replication in a clinical context.

Page 3 Introduction

“Experiments in genetically modified mice have pin-pointed this effect to incapacity of myeloid cells to limit T cell proliferation and to induce regulatory T cell (T_{reg}) activation via superoxide generation³⁵⁻³⁷. ROS has also been shown to mediate a range of immune regulatory effects including; T cell hyporesponsiveness^{38,39}, diminished T cell receptor signaling^{30,40-42}, downstream cytokine production⁴³ and T helper cell (T_H17) development⁴⁴⁻⁴⁶. In addition, memory T cells are more susceptible to ROS compared to naïve T cells and T_{reg} ⁴⁷.”

Page 9 Discussion

“So far myeloid cell-derived ROS generation as an immune regulator has received little attention during DMT intervention in RRMS, including DMF. This is despite extensive experimental data supporting a role for ROS in the regulation of adaptive immune responses in autoimmune conditions^{36,44},”

Reviewer #2

“The changes in T cell differentiation states are described in a global fashion, but it remains unclear, how what causes the relative increase in naive cells in responders.”

Action: The reviewer is correct that we used an unbiased global approach to describe changes occurring in two key cell types. Indeed, it is a descriptive approach and thus conclusions on underlying mechanisms are restricted. In order to address this comment we have revised parts of the discussion, where findings on DNA methylation in IL-6, IL-17A and IL-22 in CD4 T cells are now referred back to previous publications that are in line with our findings. In addition, we have highlighted certain specific pathways in Fig.5 and Fig.6 known to be associated with immunopathogenesis of MS.

Page 10 Discussion

“In the CD4⁺ T cells we predicted upstream regulators suggesting relevant factors for activation and function of T_H17 and T_{reg} (Supplementary Table 10). This was in line with functional implications of wide spread methylation changes validating previous published data on longitudinal changes in T_H17 and T_{reg} frequencies in man²⁶. Herein both IL-6, IL-17A and IL-22 were all differentially methylated over time in CD4⁺ T cells (Fig.5), all of them key-players in T cell subset regulation and MS pathology.

Changes in the adaptive immune cell profile were further verified with a standardized flow cytometry approach. DMF *responders* significantly increased their proportion of naïve T cells compared to *non-responders*. DMF lowered the proportion of both T_{CM} and T_{EM} cells²⁶, with the additional observation that this was true only for DMF *responders*. The lowering in proportion of T_{CM} and effector memory T cells (T_{EM}) in RRMS patients with stable disease in contrast to patients with ongoing disease has been described before²⁶. As postulated in that study, the explanation for this is likely due to migration of other T cell subsets (including T_H17) from the blood to the CNS, thus affecting the proportions T cell subset staying in the blood. This difference between *responders* and *non-responders* is further supported by the differentially methylated genes involved in leukocyte migration (*ACTINs* and *MAPKs*) (Fig.5) and plasma chemokines (CCL28) (Fig.6).”

Reviewer #2

“Fig. 5 C is poorly explained, and it is not clear, what the absolute values of cytokines are that have been measured. If the values around the graph and for each cytokine were plasma

levels, e.g. 0.3 pg/ml for IL-17A, these values would be very low and question the biological relevance of the changes.”

Action: Cytokine levels are shown as relative values as the proximity extension assay for quantifying protein levels used here, while being highly sensitive and reproducible (alike PCR), is not optimal for absolute quantifications. The fact that relative levels are presented is now indicated in the figure legend as well. The figure legend of the submitted manuscript is also referring to supplementary table 11 where all (relative) values are given. We still think that inclusion of cytokines is helpful to illustrate a pattern of changes occurring over time, with certain differences between responders and non-responders being evident. We have considered merging of the two circles, but we believe this would make the interpretation harder.

Page 8 Results

“Relative protein expression of cytokines in plasma was measured using the Olink platform at baseline and six months. This demonstrated that changes in cytokine profiles over time were more pronounced in *responders* compared to *non-responders* (Fig.6D). For example levels of IL-17C, IL-12B and CXCL9 were significantly changed in *responders* but not in *non-responders* (Supplementary Table 11).”

Page 13 Methods

“For cytokine analysis, samples were analyzed with the Immune Response Panel by Olink Proteomics, Uppsala, Sweden using the proximity extension assay for quantifying relative cytokine levels.”

Figure 6 legend

“Normalized protein expression (NPX) in plasma of IL17A, IL17C, CCL28, CD137, SLAMF7, EN-RAGE, IL12B, IL18R1, CXCL9, CCL4 and TWEAK were analyzed in *responders* (n=29) and *non-responders* (n=9) at baseline and six months after DMF intervention (D)”

Reviewer #2

“Fig 5 shows that the percentages of central- and effector memory cells differ between responders and non-responders, but this is not related to absolute lymphocyte counts, which drop substantially in DMF-treated patients.”

“As already mentioned, it is difficult to judge changes of naive and memory cell populations unless the absolute lymphocyte counts are considered.”

Action: We agree that absolute numbers, in addition to relative proportions, give additional information. The absolute lymphocyte numbers for patients from the larger cohort sampled in clinical routine is given in Fig.2G with a non-significant trend for lower lymphocyte numbers at six months for responders compared to non-responders. We now have included also absolute numbers of T cell subsets in the smaller cohort (Fig.6A-C), where both proportions and absolute T_{CM} numbers differ significantly between responders and non-responders at follow up, with a trend in the same direction also for T_{EM} cells. In contrast, proportions and absolute numbers of naïve cells at six months goes in opposite directions, suggesting that while responders have lower total lymphocyte counts, a higher proportion of these are indeed naïve cells.

Page 8 Results

“Longitudinal characterization of T cell subsets by multicolor flow cytometry showed a significant increase in proportions of naïve T cells in *responders* over time and that this

proportion was larger compared to *non-responders* at six months (Fig.6A), while total naïve T cell numbers decreased in both groups over time (Fig.6B). Proportions and absolute numbers of central memory T cell (T_{CM}) and effector memory T cells (T_{EM}) appeared to be relatively unchanged in *non-responders*, whereas these subsets were significantly lower both in relative and absolute terms in *responders* over time (Fig.6C-F).”

Reviewer #2

“For many of the measures, the numbers of samples are relatively, sometimes very small, which surprises when considering the large number of patients that have regularly been sampled. One therefore wonders how representative the observations are for the entire cohort of patients.”

“With respect to the NOX3-associated SNPs and their relation to the findings, it would be good to give the reader a sense for the number of individuals out of the entire DMF-treated cohort, which had been tested (either in the text of the figure).”

Action: We here present data on different types of blood tests from several cohorts; one smaller cohort of patients who agreed to come in for additional blood sampling for research purposes, a larger clinical cohort consisting of all patients starting DMF at our center only with lab data collected according to clinical routine, and a national cohort of patients starting DMF with genotype data available.

We are aware that this could be confusing and have now in this revised version added information in Supplementary Table 1 on every assessment performed for every subject, also comprising age, previous therapy and EDSS. This is now also graphically summarized in Fig.1C and more clearly stated in the result section. We feel it is also important to state that there was no sub-selection of subjects within each group, i.e. individuals were only selected based on the availability of samples/data with very limited and transparent censoring, e.g. subjects terminating DMF therapy at an early stage mainly due to side effects. In terms of known baseline characteristics there was a considerable overlap between the different groups, suggesting that they indeed are representative of patients starting DMF in Sweden during this period.

Page 4 Results

“We included patients from May 2014 to March 2017 starting DMF in clinical routine that volunteered for extra blood sampling, but otherwise were not subject to any other selection criteria (Fig.1A-C, Supplementary Table 1).”

Figure 1 legend

*“Peripheral blood was sampled at the clinic of patients fulfilling the criteria for RRMS and prescription of DMF. No additional selection criteria was applied. Schematic illustrating of analyzed material and assessment (A). Definition of RRMS patients as *responders* or *non-responders* to DMF therapy, applied in Fig.2 (B). Experimental overlap between assessments in the study, the detailed description is provided in Supplementary Table 1 (C).”*

Reviewer #3

-poor description of oxidative stress in the CNS especially since the ROS scavenging capacity in the CNS is higher compared to many other organs.

Action: We agree with the reviewer that redox and ROS scavenging in the CNS is an important issue, however, we would need different types of readouts than available here. Thus, it is generally accepted that relapses and contrast-enhancing brain lesions in relapsing-

remitting MS are driven by peripheral immunity, whereas disability progression in later stages (in absence of clinical or neuroradiological inflammatory disease activity) may involve redox related processes as for example reviewed by Lassmann et al. (Nat Rev Neurol. 2012 8(11):647-56) and more recently Filippi et al (Nat Rev Dis Primers. 2018 4(1):43). This could potentially be assessed by MRI techniques such as MRI spectroscopy (for glutathione or other molecules) or certain cerebrospinal fluid measures, however, unfortunately not available here. In addition, to date there is limited data to what degree DMF can pass the blood-brain barrier in humans. Thus, we have clarified that this study is focusing on events taking place in the peripheral blood, but also stressed that processes occurring in the CNS is an important field for further studies and very relevant in the context of MS brain pathology.

Page 9 Discussion

“The functional evaluation showed that DMF increased the inducible ROS generation more in *responders* than *non-responders*, which is interesting in light of the observation that NADPH oxidases have been found in active MS lesions and are believed to contribute to tissue injury⁵³. However, it is likely that the consequences of ROS generation are different depending if it triggers redox regulation or oxidative damage. Also, the site of ROS generation is likely a crucial factor since the consequences of ROS generation in the CNS parenchyma likely differ compared to secondary lymphoid organs.”

Reviewer #3

“you need to change the term “global epigenetic changes” to “DNA methylome” throughout the paper. Also in this case, there is no appropriate mention in the the Introduction of why the Authors retained relevant to study such epigenetic-related events. Thus, the Authors need to mention relevant epigenetic studies in the field (see and quote: He H et al, Hum Genomics 2018; Wang Z, et al. Curr Opin Rheumatol. 2017; Picascia A et al. J Neuroimmunol 2015) and explain why you chosen to focus your attention on DNA methylation alone.”

Action: We thank the reviewer for pointing out this important issue regarding nomenclature and have changed the text accordingly. In addition, we have added an explanation why we chose to look at DNA methylation solely.

Page 6 Results

“DNA methylation is a stable functional genomic readout able to influence transcriptional activity. DNA methylation profiling was carried out using Illumina EPIC arrays in sorted CD14⁺ monocytes from RRMS patients sampled at baseline and after three and six months. Assessment of other epigenetic changes including changes in histone modifications and non-coding RNAs were not carried out due to limitation in input material.”

Reviewer #3

“Finally, the issue of epigenetics in autoimmune disease is even more complicated (see and quote: Wu H, et al. Autoimmun Rev. 2018; Ray and Yung. Clin Immunol. 2018; Shu Y, et al. Clin Rev Allergy Immunol. 2017; Picascia A et al. Clin Immunol 2015) and, consequently, the pathogenic effects of ROS should be dissected in epigenetic and non epigenetic-sensitive mechanisms.”

Action: The assessment of DNA methylation was here used primarily as a complement to assessing mRNA expression, since our experience is that it gives a more stable functional readout from which affected functions and pathways can be inferred. The impact of inheritable epigenetic states is indeed interesting, but also very complex to address

particularly in human samples. As mentioned above, monocyte-mediated modulation of T cell activity through ROS dependent mechanisms has been shown using several different experimental approaches. This has now been added to the introduction section.

Page 3 Introduction

“However, the aim of this particular study was to perform a detailed characterization of the initial monocyte response and subsequent immunomodulation occurring in peripheral blood of RRMS patients starting therapy with DMF. In addition, changes in DNA methylation from sorted cells were used to complement and verify changes in transcription and immunoprofiling.”

Reviewer #3

Revise authors conflicts of interest

Action: This has been updated.

Competing interests

“FP has received research grants from Biogen, Novartis, and Genzyme, and fees for serving as Chair of DMC in clinical trials with Parexel.

TO has received unrestricted MS research grants, and/or lecture advisory board honoraria from Biogen, Novartis, Sanofi and Roche.”

Reviewer #3

Poorly written abstract

Action: The abstract has been revised, with clarification of background and scientific rationale for the study.

Reviewer #3

“Delete Panel B of Figure 2.”

Action: With respect for the reviewer’s opinion, we would argue that the differences between longitudinal DNA methylation changes in CD14+ and CD4+ cells is a central finding here, and that omission of panel B would lead to loss of information.

Reviewer #3

Revise reference list regarding numbering.

Action: This has been revised

Reviewer #3

“Thus, in the setting of isolated monocytes from patients treated with DMF, additional experiments should be carried to better understand the effect of ROS in the presence of specific scavengers and/or substances inhibitors of ROS”

Action:

This is a very appealing suggestion and a good hypothesis for following studies. Recent articles with a similar approach as suggested have shed light on regulation of T cell proliferation and activity in MS (*PubMedID:30173916*). However there is also the risk of inducing artifacts when studying these cells *ex vivo*. A body of experimental data suggest

myeloid cell-derived ROS to be able to play a regulatory role on T cell activity (PubMedID:10799876, 20861446, 21593419, 18771940, 20881184, 20682913, 20682913, 18383034, 15310853) however conclusive characterization of ROS generation in human autoimmune patients during therapy is lacking. Thus, the main objective with this study was to address these *in vivo* ROS-mediated mechanisms in man. Herein we can show that alternation of methylation in NOX3 promotor region is suggestively associated with SNP in within the gene and that this affects the transcriptional levels of NOX3 as a likely explanation for the altered ROS generation observed in monocytes.

REVIEWERS' COMMENTS:

Reviewer #2 (Remarks to the Author):

The manuscript of Carlström and colleagues has been improved considerably by the changes, corrections and explanations to the reviewers' remarks. Still, there are inconsistencies in the data, and the description of the results and their interpretations could be improved. There are still many statements that are difficult to understand:

Examples:

"However, since Nrf2 is a sensor for oxidative stress, its activation by DMF may also include transient oxidative responses. There are also discrepancies in the literature how to assess oxidative stress and whether fluctuations of anti-oxidative transcripts..." What do the authors want to say here? The observation that Nrf2 participates in sensing oxidative stress does not necessarily imply that it induces/includes (?) oxidative responses.

"...net increase in oxidative environment" What is meant here?

See also later: "The recorded changes in monocyte transcription and methylation could result both from changes causing elevated oxidative stress, but also consequences of an increased oxidative environment." What do the authors want to say here?

"Lastly, we identify a single nucleotide polymorphism (SNP) in the NOX3 gene to be associated to treatment response to DMF and suggestively associated with reduced ROS generation." This sentence is hard to grasp in the context of the overall findings. Is the suggestion of the authors that the DMF response-related SNP in the NOX3 gene leads to a loss of function with respect to ROS production? Their data indicate that the response to DMF is related to increased isoprostane release and ROS production, and that the latter, based on existing evidence, is consistent with the (T cell) immunomodulatory effects of ROS. In animal models, knock out of NOX genes and reduced production of ROS are related to increased severity of autoimmune disease models. The above sentence would suggest the opposite. This needs to be clarified/re-worded.

The words "suggested, suggestively" are being used a lot and convey a high degree of vagueness. As examples, see:

"The suggestively lower degree of CpG methylation in responders with the A allele at rs6919626 was further reflected in higher NOX3 transcription in CD14+ monocytes at 6 months (Fig.4D). Together, this suggests that genetic variation and CpG methylation in monocytic NOX3 might influence the NOX3 gene transcription and thus monocyte function." Together with the fact that the sentences are difficult to understand, the repeated use of suggest and might makes the reader wonder what the authors think about their own data.

See also: "We herein provide novel insights into the early effects of DMF on monocytes that are suggested to be associated with modulation of downstream adaptive immune responses."

In the above sentence and at several other points the words upstream and downstream, which are easily understood in the context of genetics and positions of genes, are misleading and uncommon in their use in an immunological context. The authors want to infer cause and effect relationships in immune responses, e.g. between monocytes and T cells, but it is in most cases not clear what they want to say when using the words upstream and downstream.

See e.g.: "The notion of monocytes being an upstream target of DMF is also supported..."

Some of the aspects regarding DMF effects on monocytes and dependent effects on T cells could be studied entirely in vitro or with monocytes ex vivo isolated at the different time points from patients under DMF therapy and responding or not. Such data would strengthen many of the speculative conclusions substantially.

"Herein both IL-6, IL-17A and IL-22 were all differentially methylated over time in CD4+ T cells (Fig.5), all of them key-players in T cell subset regulation and MS pathology." One should be more cautious with statements like this. Based on current data, one would not call IL-6, IL-17A and IL-22 key players in MS. There is some data indicating that each of these molecules may be important in MS, but this evidence is not sufficient to label them key players.

Among the many possibilities how DMF-treated monocytes could directly or indirectly via their influences on T cells act, i.e. regulatory effects, influence on migration, apoptosis, etc., which the authors all cite or mention, it would be good if they weighed their evidence and tried to come to concise conclusions vis-a-vis the published evidence. Again, it would help to support statements by more mechanistic data.

See for example: "The lowering in proportion of TCM and effector memory T cells (TEM) in RRMS patients with stable disease in contrast to patients with ongoing disease has been described before²⁶. As postulated in that study, the explanation for this is likely due to migration of other T cell subsets (including TH17) from the blood to the CNS, thus affecting the proportions T cell subset staying in the blood." It seems that the authors conclude from the reduced numbers of TCM and TEM that these cells have migrated to the brain, which makes little sense since the reduction is found primarily in individuals who respond to DMF. Disease-relevant T cells should be kept out of the CNS compartment.

Monocyte counts: In Fig. 2B, the authors show that the monocyte counts in untreated patients overall are approximately 0.8×10^3 (per μl ; one has to assume). If one adds the numbers for the DMF-treated patients (responders) for the three types of monocytes, the numbers are approximately 11×10^3 (per μl). These monocyte numbers cannot be correct and should be checked. The correct units should be shown as well, i.e. cells per what volume. In Fig. 6, for instance, cell numbers $\times 10^9/\text{l}$ is used.

Reviewer #3 (Remarks to the Author):

The revised manuscript appears improved in comparison to the original submission. However, many issues have not addressed satisfactorily throughout the revised manuscript.

- 1) As indicated, Authors acknowledged that MRI-derived techniques such as MRI spectroscopy (for glutathione or other molecules) and other nuclear medicine techniques could improve the findings of the study. However, they did not mention these techniques in the revised version. Moreover, the kinetic of DMF in the human blood-brain barrier may help to understand the clinical relevance of the proposed study. I recommend to mention these limitations in the revised manuscript.
- 2) I have appreciated the explanation regarding the limitation of the investigation to DNA methylation (ie, limitation of the input material). However, Authors need to clarify that important additional information will be added by future (possible) studies regarding miRNA and histone-related modifications in order to have a complete picture of epigenetic sensitive events (again see and try to quote: He H et al, Hum Genomics 2018; Wang Z, et al. Curr Opin Rheumatol. 2017; Picascia A et al. J Neuroimmunol 2015).
- 3) As stated by the Authors "In addition, changes in DNA methylation from sorted cells were used to complement and verify changes in transcription and immunoprofiling." This is NOT necessarily true. Epigenetic mechanisms may influence a cascade of events beyond the "complimentary" use done in study design proposed by Authors. Increasing pathogenic evidence remarks the concept that such

epigenetic events may influence per se both immune response and clinical degree of autoimmune disease (again see and try to quote: Wu H, et al. *Autoimmun Rev.* 2018; Ray and Yung. *Clin Immunol.* 2018; Shu Y, et al. *Clin Rev Allergy Immunol.* 2017; Picascia A et al. *Clin Immunol* 2015).

4) The abstract is still poorly written (even considered the General Reader of the Journal). Some vague statements should be replaced. For example: "peripheral blood cells" should be replaced by exact cell types; "Monocytes change in their numbers" states clearly if increased or decreased; "changes are evident in T cells" evident should be replaced by numbers (and possibly significance); "NOX3 gene is associated" which kind of association here? Positive or negative? R value?; "this provide novel information on the mechanisms of DMF and the role of monocytes in autoimmune diseases" which information? Which Role? I recommend to amend Abstract with missing informations before its publication.

5) The reply regarding the possible addition of additional experiments with ROS scavengers is not well addressed. The possibility to suffer of inducing artifacts in ex vivo studies is too vague and general. All in vitro studies may suffer of such limitations. It would be useful to use physiological concentrations (to mime condition closer to humans) of such scavengers and dose-response curves. Again, the reduced availability of human blood samples is a more reasonable explanation here.

Reviewer #2 (Remarks to the Author):

The manuscript of Carlström and colleagues has been improved considerably by the changes, corrections and explanations to the reviewers' remarks. Still, there are inconsistencies in the data, and the description of the results and their interpretations could be improved. There are still many statements that are difficult to understand:

Examples:

"However, since Nrf2 is a sensor for oxidative stress, it's activation by DMF may also include transient oxidative responses. There are also discrepancies in the literature how to assess oxidative stress and whether fluctuations of anti-oxidative transcripts..." What do the authors want to say here? The observation that Nrf2 participates in sensing oxidative stress does not necessarily imply that it induces/includes (?) oxidative responses.

"...net increase in oxidative environment" What is meant here?

See also later: "The recorded changes in monocyte transcription and methylation could result both from changes causing elevated oxidative stress, but also consequences of an increased oxidative environment." What do the authors want to say here?

"Lastly, we identify a single nucleotide polymorphism (SNP) in the NOX3 gene to be associated to treatment response to DMF and suggestively associated with reduced ROS generation." This sentence is hard to grasp in the context of the overall findings. Is the suggestion of the authors that the DMF response-related SNP in the NOX3 gene leads to a loss of function with respect to ROS production? Their data indicate that the response to DMF is related to increased isoprostane release and ROS production, and that the latter, based on existing evidence, is consistent with the (T cell) immunomodulatory effects of ROS. In animal models, knock out of NOX genes and reduced production of ROS are related to increased severity of autoimmune disease models. The above sentence would suggest the opposite. This needs to be clarified/re-worded.

The words "suggested, suggestively" are being used a lot and convey a high degree of vagueness. As examples, see:

"The suggestively lower degree of CpG methylation in responders with the A allele at rs6919626 was further reflected in higher NOX3 transcription in CD14+ monocytes at 6 months (Fig.4D). Together, this suggests that genetic variation and CpG methylation in monocytic NOX3 might influence the NOX3 gene transcription and thus monocyte function." Together with the fact that the sentences are difficult to understand, the repeated use of suggest and might makes the reader wonder what the authors think about their own data.

See also: "We herein provide novel insights into the early effects of DMF on monocytes that are suggested to be associated with modulation of downstream adaptive immune responses."

In the above sentence and at several other points the words upstream and downstream, which are easily understood in the context of genetics and positions of genes, are misleading and uncommon in their use in an immunological context. The authors want to infer cause and effect relationships in immune responses, e.g. between monocytes and T cells, but it is in most cases not clear what they want to say when using the words upstream and downstream.

See e.g.: "The notion of monocytes being an upstream target of DMF is also supported..."

Some of the aspects regarding DMF effects on monocytes and dependent effects on T cells could be studied entirely in vitro or with monocytes ex vivo isolated at the different time points from patients under DMF therapy and responding or not. Such data would strengthen many of the speculative conclusions substantially.

"Herein both IL-6, IL-17A and IL-22 were all differentially methylated over time in CD4+ T cells (Fig.5), all of them key-players in T cell subset regulation and MS pathology." One should be more cautious with statements like this. Based on current data, one would not call IL-6, IL-17A and IL-22 key players in MS. There is some data indicating that each of these molecules may be important in MS, but this evidence is not sufficient to label them key players.

Among the many possibilities how DMF-treated monocytes could directly or indirectly via their influences on T cells act, i.e. regulatory effects, influence on migration, apoptosis, etc., which the authors all cite or mention, it would be good if they weighed their evidence and tried to come to concise conclusions vis-a-vis the published evidence. Again, it would help to support statements by more mechanistic data.

See for example: "The lowering in proportion of TCM and effector memory T cells (TEM) in RRMS patients with stable disease in contrast to patients with ongoing disease has been described before²⁶. As postulated in that study, the explanation for this is likely due to migration of other T cell subsets (including TH17) from the blood to the CNS, thus affecting the proportions T cell subset staying in the blood." It seems that the authors conclude from the reduced numbers of TCM and TEM that these cells have migrated to the brain, which makes little sense since the reduction is found primarily in individuals who respond to DMF. Disease-relevant T cells should be kept out of the CNS compartment.

Monocyte counts: In Fig. 2B, the authors show that the monocyte counts in untreated patients overall are approximately 0.8×10^3 (per μl ; one has to assume). If one adds the numbers for the DMF-treated patients (responders) for the three types of monocytes, the numbers are approximately 11×10^3 (per μl). These monocyte numbers cannot be correct and should be checked. The correct units should be shown as well, i.e. cells per what volume. In Fig. 6, for instance, cell numbers $\times 10^9/\text{l}$ is used.

Reviewer #3 (Remarks to the Author):

The revised manuscript appears improved in comparison to the original submission. However, many issues have not addressed satisfactorily throughout the revised manuscript.

- 1) As indicated, Authors acknowledged that MRI-derived techniques such as MRI spectroscopy (for glutathione or other molecules) and other nuclear medicine techniques could improve the findings of the study. However, they did not mention these techniques in the revised version. Moreover, the kinetic of DMF in the human blood-brain barrier may help to understand the clinical relevance of the proposed study. I recommend to mention these limitations in the revised manuscript.
- 2) I have appreciated the explanation regarding the limitation of the investigation to DNA methylation (ie, limitation of the input material). However, Authors need to clarify that important additional information will be added by future (possible) studies regarding miRNA and histone-related modifications in order to have a complete picture of epigenetic sensitive events (again see and try to quote: He H et al, Hum Genomics 2018; Wang Z, et al. Curr Opin Rheumatol. 2017; Picascia A et al. J Neuroimmunol 2015).
- 3) As stated by the Authors "In addition, changes in DNA methylation from sorted cells were used to complement and verify changes in transcription and immunoprofiling." This is NOT necessarily true. Epigenetic mechanisms may influence a cascade of events beyond the "complimentary" use done in study design proposed by Authors. Increasing pathogenic evidence remarks the concept that such epigenetic events may influence per se both immune response and clinical degree of autoimmune disease (again see and try to quote: Wu H, et al. Autoimmun Rev. 2018; Ray and Yung. Clin Immunol. 2018; Shu Y, et al. Clin Rev Allergy Immunol. 2017; Picascia A et al. Clin Immunol 2015).
- 4) The abstract is still poorly written (even considered the General Reader of the Journal). Some vague statements should be replaced. For example: "peripheral blood cells" should be replaced by exact cell types; "Monocytes change in their numbers" states clearly if increased or decreased; "changes are evident in T cells" evident should be replaced by numbers (and possibly significance); "NOX3 gene is associated" which kind of association here? Positive or negative? R value?; "this provide novel information on the mechanisms of DMF and the role of monocytes in autoimmune diseases" which information? Which Role? I recommend to amend Abstract with missing informations before its publication.
- 5) The reply regarding the possible addition of additional experiments with ROS scavengers is not well addressed. The possibility to suffer of inducing artifacts in ex vivo studies is too vague and general. All in vitro studies may suffer of such limitations. It would be useful to use physiological concentrations (to mime condition closer to humans) of such scavengers and dose-response curves. Again, the reduced availability of human blood samples is a more reasonable explanation here.

Point by point critique

Essential from the editor.

As you will see, they emphasize the need to improve the clarity of the manuscript text and data presentation. My recommendation is to ask your colleagues who are not directly involved in the study to read it carefully, as if they were to repeat and interpret the experiments, so that they would flag any aspects that are unclear.

Reviewer #2, expertise in clinical multiple sclerosis

There are still many statements that are difficult to understand:

"However, since Nrf2 is a sensor for oxidative stress, its activation by DMF may also include transient oxidative responses. There are also discrepancies in the literature how to assess oxidative stress and whether fluctuations of anti-oxidative transcripts..." What do the authors want to say here? The observation that Nrf2 participates in sensing oxidative stress does not necessarily imply that it induces/includes (?) oxidative responses.

ANSWER We agree with the reviewer that this statement is unclear and it has been changed and moved accordingly

Page 4 Results

"Peripheral blood was collected at regular intervals before (baseline) and during the first six months after starting DMF and patients were followed in order to evaluate treatment efficacy according to clinical routine. To address oxidative stress, we initially determined plasma levels of free 8.12-iso-iPF2 α -VI isoprostane generated through non-enzymatic oxidation, considered as the most acknowledged technique of quantifying net oxidative stress^{14,15} and superior to measurement of e.g. anti-oxidative enzymes since this technique measure oxidation instead of secondary responses to oxidation"

"...net increase in oxidative environment" What is meant here?

ANSWER Our intention with the use of "net" is to include the fact that chemically reactions of reduction and oxidation occurs simultaneously. However the use of just "increase" may be as suitable and we have thus changed the it to "increase"

"The recorded changes in monocyte transcription and methylation could result both from changes causing elevated oxidative stress, but also consequences of an increased oxidative environment." What do the authors want to say here?

ANSWER This statement is a bit redundant and misleading and has thus been removed

"Lastly, we identify a single nucleotide polymorphism (SNP) in the NOX3 gene to be associated to treatment response to DMF and suggestively associated with reduced ROS generation." This sentence is hard to grasp in the context of the overall findings. Is the suggestion of the authors that the DMF response-related SNP in the NOX3 gene leads to a loss of function with respect to ROS production? Their data indicate that the response to DMF is related to increased isoprostane release and ROS production, and that the latter, based on existing evidence, is consistent with the (T cell) immunomodulatory effects of ROS. In animal

models, knock out of NOX genes and reduced production of ROS are related to increased severity of autoimmune disease models. The above sentence would suggest the opposite. This needs to be clarified/re-worded.

ANSWER The reviewer is correct, and the animal studies the reviewer is referring to are also the foundation for this study. "Treatment response to DMF" does not say if it's a beneficial response or not. However, in the second part we state the direction of the association "reduced ROS generation". To make this consistent and clearer we thus have rephrased this statement, beneficial treatment response to DMF and the same allele is associated with increased ROS generation.

Page 4 Results

"Lastly, we identify a single nucleotide polymorphism (SNP) in the NOX3 gene to be associated to a beneficial treatment response to DMF and suggestively associated with increased ROS generation."

The words "suggested, suggestively" are being used a lot and convey a high degree of vagueness. As examples, see:

"The suggestively lower degree of CpG methylation in responders with the A allele at rs6919626 was further reflected in higher NOX3 transcription in CD14+ monocytes at 6 months (Fig.4D). Together, this suggests that genetic variation and CpG methylation in monocytic NOX3 might influence the NOX3 gene transcription and thus monocyte function." Together with the fact that the sentences are difficult to understand, the repeated use of suggest and might makes the reader wonder what the authors think about their own data.

ANSWER The data of these statements is not significant and thus we used this wording avoiding to overstate our findings. However, we do agree that this make the statement more vague than necessary. In the previous sentence, we have already declared that the findings are not significant thus we find it better and clearer to rephrase the section as followed.

Page 7 Results

"Reduced CpG methylation in DMF responders carrying the A allele could be further linked to higher NOX3 transcription in CD14⁺ monocytes at 6 months (Fig.4D). Together, this finding suggests that genetic variation and CpG methylation in monocytic NOX3 might influence the NOX3 gene transcription and thus monocyte function, particularly ROS production."

See also: "We herein provide novel insights into the early effects of DMF on monocytes that are suggested to be associated with modulation of downstream adaptive immune responses."

ANSWER This statement has been rephrased

Page 9 Discussion

"We herein provide novel insights into early effects of DMF on monocytes that could be of importance for subsequent modulation of adaptive immune responses."

See e.g.: "The notion of monocytes being an upstream target of DMF is also supported..."

ANSWER This statement has been rephrased

Page 10/11 Discussion

"Our hypothesis of monocytes being primarily affected by DMF is also supported by the temporal profile of methylation changes in a smaller sample set, where changes in CD14⁺ monocytes occurred prior to changes in CD4⁺ T cells. Moreover, changes in monocyte numbers occurring prior to changes in lymphocyte numbers in the larger clinical cohort further support monocytes being targeted by DMF prior to CD4⁺ T cells."

In the above sentence and at several other points the words upstream and downstream, which are easily understood in the context of genetics and positions of genes, are misleading and uncommon in their use in an immunological context. The authors want to infer cause and effect relationships in immune responses, e.g. between monocytes and T cells, but it is in most cases not clear what they want to say when using the words upstream and downstream.

ANSWER The authors agree with the reviewer's statement. The term "upstream/downstream" is throughout used both in a genetic context, in a longitudinal context but also to describe the cells in a signaling hierarchy. The use of "upstream/downstream" has been revised throughout and is now only used in a genetic context. See the examples above.

Some of the aspects regarding DMF effects on monocytes and dependent effects on T cells could be studied entirely in vitro or with monocytes ex vivo isolated at the different time points from patients under DMF therapy and responding or not. Such data would strengthen many of the speculative conclusions substantially.

ANSWER The authors are aware of this and agree with the reviewer. This is one aspect that one could build on in coming studies. However, there is a substantial body of experimental evidence describing the effects of (myeloid) ROS on T cell function. Thus our focus with this particular study was to evaluate the in vivo situation focusing on role of ROS. For example, in Fig3 and Fig5 we have grouped changes in genes based on the involvement of ROS associated genes (defined in the method section). These datasets provides novel insight and could, in our opinion not be replaced with ex vivo experiments. However, we agree that some of the findings described in our study would be worth exploring in other (experimental) settings in future studies.

"Herein both IL-6, IL-17A and IL-22 were all differentially methylated over time in CD4⁺ T cells (Fig.5), all of them key-players in T cell subset regulation and MS pathology." One should be more cautious with statements like this. Based on current data, one would not call IL-6, IL-17A and IL-22 key players in MS. There is some data indicating that each of these molecules may be important in MS, but this evidence is not sufficient to label them key players.

ANSWER The authors agree with the reviewer's comment and have rephrased the statement accordingly

Page 11 Discussion

“Herein both IL-6, IL-17A and IL-22 were all differentially methylated over time in CD4⁺ T cells (Fig.5), all of which contribute to T cell subset regulation and MS pathology.”

Among the many possibilities how DMF-treated monocytes could directly or indirectly via their influences on T cells act, i.e. regulatory effects, influence on migration, apoptosis, etc., which the authors all cite or mention, it would be good if they weighed their evidence and tried to come to concise conclusions vis-a-vis the published evidence. Again, it would help to support statements by more mechanistic data.

ANSWER The authors agree with the reviewer’s comment and have rephrased the statement accordingly

Page 11 Discussion

“The absolute number of naïve T cells did decrease in both groups, suggesting that absolute number of naïve T cells is insufficient to predict beneficial treatment outcome. Lowering of T_{CM} and T_{EM} in RRMS patients with ongoing disease has been described before²⁵. Our data further implies that reduction of T_{CM} could be relevant for beneficial DMF response. The significant difference in T_{CM} number between *responders* and *non-responders* could further be supported by the differentially methylated genes involved in T cell differentiation, clonal expansion (Hippo) and T cell apoptosis (Fig.5C). Interestingly, these pathways were also highly influenced by oxidative stress. Pathways involved in lymphocyte trafficking and migration are also changed over time and between *responders* and *non-responders* (Fig.5C, Fig.6G). This pathways could be of importance in e.g. T_{H17} and T_{reg} migrating to the CNS, leaving the blood. Since also T_{CM} and T_{EM} in the CNS are associated with disease progression, and are decreasing in blood of *responders*, CNS migratory pathways are likely not of relevance in T_{CM} and T_{EM} in our cohort. Decreased migration would suggestively cause an accumulation in the blood²⁵.”

See for example: “The lowering in proportion of TCM and effector memory T cells (TEM) in RRMS patients with stable disease in contrast to patients with ongoing disease has been described before²⁶. As postulated in that study, the explanation for this is likely due to migration of other T cell subsets (including TH17) from the blood to the CNS, thus affecting the proportions T cell subset staying in the blood.” It seems that the authors conclude from the reduced numbers of TCM and TEM that these cells have migrated to the brain, which makes little sense since the reduction is found primarily in individuals who respond to DMF. Disease-relevant T cells should be kept out of the CNS compartment.

ANSWER Our intention was not to state that T_{EM} and T_{CM} have migrated to the CNS or that the migration of these cell types to the CNS is the reason why responders and non-responders show different levels of these cells in peripheral blood. We apologize for the misleading phrasing. The statement has now been changed.

Page 11 Discussion

“The absolute number of naïve T cells did decrease in both groups, suggesting that absolute number of naïve T cells is insufficient to predict beneficial treatment outcome. Lowering of T_{CM} and T_{EM} in RRMS patients with ongoing disease has been described before²⁵. Our data further implies that reduction of T_{CM} could be relevant for beneficial DMF response. The significant difference in T_{CM} number between *responders* and *non-*

responders could further be supported by the differentially methylated genes involved in T cell differentiation, clonal expansion (Hippo) and T cell apoptosis (Fig.5C). Interestingly, these pathways were also highly influenced by oxidative stress. Pathways involved in lymphocyte trafficking and migration are also changed over time and between *responders* and *non-responders* (Fig.5C, Fig.6G). This pathways could be of importance in e.g. T_H17 and T_{reg} migrating to the CNS, leaving the blood. Since also T_{CM} and T_{EM} in the CNS are associated with disease progression, and are decreasing in blood of *responders*, CNS migratory pathways are likely not of relevance in T_{CM} and T_{EM} in our cohort. Decreased migration would suggestively cause an accumulation in the blood²⁵."

Monocyte counts: In Fig. 2B, the authors show that the monocyte counts in untreated patients overall are approximately 0.8 x 10³ (per µl; one has to assume). If one adds the numbers for the DMF-treated patients (responders) for the three types of monocytes, the numbers are approximately 11 x 10³ (per µl). These monocyte numbers cannot be correct and should be checked. The correct units should be shown as well, i.e. cells per what volume. In Fig. 6, for instance, cell numbers x 10⁹/l is used.

ANSWER Fig2B and Fig2C-E depicts data from the same patients from the same blood sample. However, they have been analyzed with two different protocols which explains the discrepancies in cell number. We don't have any data on CD14/CD16 distribution in healthy controls (Fig2C-E). The data in Fig2B is very clear however we were not able to validate it in the larger cohort shown in Fig2F. One reason for this might be that data in Fig2F-G is blood cell count derived from the clinic and not analyzed by flow cytometry (this is further illustrated in Fig.1C and supplementary Table 1. Our suggestion is thus to remove Fig2B since we can't verify this data with a different technique, despite a clear primary finding.

Commented [KC1]: Delete it or clarify in text?

Regarding the volume cells in Fig2B-E, H-I has been acquired by time with flow cytometry meanwhile Fig2F-G is data generated from the clinical routine data. This should be stated more clearly in the method section.

“All samples were analyzed with a 3 laser Beckman Coulter Gallios using Kaluza Software (Beckman Coulter, Brea, CA) and acquired by time.”

Reviewer #3, expertise in ROS and epigenetics of autoimmune

1.As indicated, Authors acknowledged that MRI-derived techniques such as MRI spectroscopy (for glutathione or other molecules) and other nuclear medicine techniques could improve the findings of the study. However, they did not mention these techniques in the revised version. Moreover, the kinetic of DMF in the human blood-brain barrier may help to understand the clinical relevance of the proposed study. I recommend to mention these limitations in the revised manuscript.

ANSWER: We state on page 4 in the introduction, that prior to measurement of oxidized isoprostanes different anti-oxidative enzymes (including GSH) were used to evaluate oxidative stress. The obvious shortcoming of using these secondary effects to evaluate oxidative stress whether low levels of anti-oxidative enzymes reflects depletion due to high oxidative stress or are low due to the absence of oxidative stress. This applies even if you evaluate GSH with MRI. The authors however agree with the reviewer that the kinetic of DMF in the CNS is still poorly understood and also state this on page 3 in the introduction. The characterization of DMF CNS kinetics using MRI is not within the scope of this project. However, the authors agree that that kind of data likely would provide good evidence if the main effect of DMF is carried out in the CNS or in the blood.

2.I have appreciated the explanation regarding the limitation of the investigation to DNA methylation (ie, limitation of the input material). However, Authors need to clarify that important additional information will be added by future (possible) studies regarding miRNA and histone-related modifications in order to have a complete picture of epigenetic sensitive events (again see and try to quote: He H et al, Hum Genomics 2018; Wang Z, et al. Curr Opin Rheumatol. 2017; Picascia A et al. J Neuroimmunol 2015).

ANSWER This has now been clarified.

Page 5 Results

“Differentially expressed genes in CD14⁺ monocytes (Supplementary Table 2) together with canonical pathways (Fig.1G, Supplementary Table 3) implicated signaling pathways related to ROS generation and functions previously ascribed to effects mediated by DMF.”

Page 10 Discussion

“DMF-dependent CNS migration of myeloid cells has been described⁵⁹ and is further supported by our longitudinal DNA methylation changes in relevant pathways involved in migration and communication (Fig.3). However, given the complex interaction between several epigenetic modalities in the regulation of transcription and cellular functions, addressing histone modifications and non-coding RNAs might provide additional insight into DMF-mediated effect at the molecular level. The functional evaluation showed that DMF increased the inducible ROS generation more in responders than non-responders,

which is interesting in light of the observation that NADPH oxidases have been found in active MS lesions and are believed to contribute to tissue injury⁵²”

Page 15 Results

“CD4⁺ and CD14⁺ cells were prepared within an hour after sampling using AutoMACS (Milteny Biotec, Bergisch, Germany) due to the manufactures standard protocol and stored in -70°C. DNA and RNA was extracted simultaneously using Qiagen Allprep DNA/RNA kit (Qiagen, Venlo, Netherlands) due to the manufactures standard protocol and stored in -70°C. This kit does not allow extraction of short RNAs.”

3. As stated by the Authors “In addition, changes in DNA methylation from sorted cells were used to complement and verify changes in transcription and immunoprofiling.” This is NOT necessarily true. Epigenetic mechanisms may influence a cascade of events beyond the “complimentary” use done in study design proposed by Authors. Increasing pathogenic evidence remarks the concept that such epigenetic events may influence per se both immune response and clinical degree of autoimmune disease (again see and try to quote: Wu H, et al. Autoimmun Rev. 2018; Ray and Yung. Clin Immunol. 2018; Shu Y, et al. Clin Rev Allergy Immunol. 2017; Picascia A et al. Clin Immunol 2015).

ANSWER MAJA?

Commented [LK2]: I sense you *have* to quote one of these articles (2nd reminder from the reviewer)

4. The abstract is still poorly written (even considered the General Reader of the Journal). Some vague statements should be replaced. For example: “peripheral blood cells” should be replaced by exact cell types; “Monocytes change in their numbers” states clearly if increased or decreased; “changes are evident in T cells” evident should be replaced by numbers (and possibly significance); “NOX3 gene is associated” which kind of association here? Positive or negative? R value?; “this provide novel information on the mechanisms of DMF and the role of monocytes in autoimmune diseases” which information? Which Role? I recommend to amend Abstract with missing informations before its publication.

ANSWER The authors agree that some info could be added to the abstract to make it even more detailed. Now cell types and direction of correlation has been added.

Commented [LK3]: I kinda agree with the reviewer.

Page 2 Abstract

“Dimethyl fumarate (DMF) is a first-line-treatment for relapsing-remitting multiple sclerosis (RRMS) having a delayed therapeutic action on T cells. The transcription factor Nrf2 essential in redox balance is the main target of DMF, but the precise therapeutic mechanisms in human remain elusive. We herein longitudinally assessed monocytes and T cells in RRMS patients to identify such mechanisms. DMF increased oxidative environment in blood. Importantly, monocytic ROS generation is maintained in patients with beneficial treatment-response in comparison to patients with disease breakthrough. Most of the changes, including methylome and transcriptome profiles, occur in monocytes prior to T cells. A single nucleotide polymorphism (SNP) in the ROS-generating NOX3 gene is associated both with beneficial DMF treatment-response and increased ROS generation. This is the first genetic association to treatment reported for DMF. This provide novel information on the mechanisms of DMF and the role of monocytes-derived oxidative processes in autoimmune diseases and their treatment.”

5. The reply regarding the possible addition of additional experiments with ROS scavengers is not well addressed. The possibility to suffer of inducing artifacts in ex vivo studies is too vague and general. All in vitro studies may suffer of such limitations. It would be useful to use physiological concentrations (to mime condition closer to humans) of such scavengers and dose-response curves. Again, the reduced availability of human blood samples is a more reasonable explanation here.

ANSWER We agree with the reviewer. The addition of further ex vivo or in vitro studies would perhaps generate additional information on the interplay between monocytes and T cells. However, there is a substantial body of experimental evidence describing the effects of (myeloid) ROS on T cell function. To verify the genetic findings in NOX3 (n=116) by doing ex vivo or in vitro experiments over time is a captivating idea but would also be a major endeavor, not necessarily reflecting true in vivo conditions. Thus we welcoming future studies to strengthen our initial and novel findings that we have based on a large body of experimental evidence.

Reviewer #1, expertise in human monocytes